# Elucidation of unusual biosynthesis and DnaN-targeting mode of action of potent anti-tuberculosis antibiotics Mycoplanecins

Chengzhang Fu [1,2,7], Yunkun Liu[1,7], Christine Walt[1,3,7], Sari Rasheed[1,3,7], Chantal D. Bader[1,3], Peer Lukat[4], Markus Neuber[1,3], F. P. Jake Haeckl[1,3], Wulf Blankenfeldt [4], Olga V. Kalinina [5,6] & Rolf Müller [1,2,3]✉

DNA polymerase III sliding clamp (DnaN) was recently validated as a new anti-tuberculosis target employing griselimycins. Three $(2S,4R)$−4-methylproline moieties of methylgriselimycin play significant roles in target binding and metabolic stability. Here, we identify the mycoplanecin biosynthetic gene cluster by genome mining using bait genes from the 4-methylproline pathway. We isolate and structurally elucidate four mycoplanecins comprising scarce homo-amino acids and 4-alkylprolines. Evaluating mycoplanecin E against *Mycobacterium tuberculosis* surprisingly reveals an excitingly low minimum inhibition concentration at 83 ng/mL, thus outcompeting griselimycin by approximately 24-fold. We show that mycoplanecins bind DnaN with nanomolar affinity and provide a co-crystal structure of mycoplanecin A-bound DnaN. Additionally, we reconstitute the biosyntheses of the unusual L-homoleucine, L-homonorleucine, and $(2S,4R)$−4-ethylproline building blocks by characterizing in vitro the full set of eight enzymes involved. The biosynthetic study, bioactivity evaluation, and drug target validation of mycoplanecins pave the way for their further development to tackle multidrug-resistant mycobacterial infections.

Tuberculosis (TB) is a severe infectious disease caused by the pathogenic bacterium *Mycobacterium tuberculosis* (*Mtb*). TB remains an enormous global health burden, causing an estimated 1.5 million deaths and 10 million new cases in 2020. TB was the leading cause of death in all infectious diseases until the coronavirus (COVID-19) pandemic. Additionally, the growing percentage of multidrug-resistant (MDR) TB is a daunting obstacle to global TB treatment and prevention efforts. Since MDR TB is resistant to at least the two most potent TB drugs, isoniazid and rifampin, patients need new treatment options[1,2].

Consequently, new drugs addressing novel targets in *Mtb* are increasingly desired.

Griselimycins (GMs) are such promising anti-TB antibiotics that bind and inhibit the mycobacterial DNA polymerase III sliding clamp (DnaN). This unique target makes GM distinct from other antibiotics to evade common TB drug resistance[3]. The unusual nonproteogenic amino acid $(2S,4R)$−4-methylproline $((2S,4R)$−4-MePro) acts as an essential building block occurring twice in GM and thrice in methylgriselimycin (MGM). Oxidation of the metabolically unstable Pro8

[1]Helmholtz Institute for Pharmaceutical Research Saarland (HIPS), Helmholtz Centre for Infection Research (HZI), and Department of Pharmacy, Saarland University, 66123 Saarbrücken, Germany. [2]Helmholtz International Lab for Anti-Infectives, Helmholtz Center for Infection Research, 38124 Braunschweig, Germany. [3]German Centre for Infection Research (DZIF), 38124 Braunschweig, Germany. [4]Structure and Function of Proteins, Helmholtz Centre for Infection Research, Inhoffenstr. 7, 38124 Braunschweig, Germany. [5]Medical Faculty, Saarland University, 66421 Homburg, Germany. [6]Helmholtz Institute for Pharmaceutical Research Saarland (HIPS), Helmholtz Centre for Infection Research (HZI), and Center for Bioinformatics, Saarland Informatics Campus, 66123 Saarbrücken, Germany. [7]These authors contributed equally: Chengzhang Fu, Yunkun Liu, Christine Walt, Sari Rasheed. ✉e-mail: rolf.mueller@helmholtz-hips.de

residue initiates GM degradation. The significantly improved metabolic stability of GM derivatives with substituent groups at Pro8, such as the natural MGM with (2 S,4 R)−4-MePro, has corroborated this finding[3]. More importantly, GM congeners with different 4-alkyl groups show improved activity against *Mtb* which might be attributed to their enhanced lipophilicity[3]. In our previous study, the Fe(II)/α-ketoglutarate (α-KG)-dependent dioxygenase GriE was proven to initiate the biosynthesis of (2 S,4 R)−4-MePro by hydroxylating L-leucine to form (2 S,4 R)−5-hydroxyleucine. The zinc-dependent dehydrogenase GriF then converts (2 S,4 R)−5-hydroxyleucine to (2 S,4 R)−4-methylglutamate-5-semialdehyde. The resulting semialdehyde will be turned to (2 S,4 R)−4-MePro by a spontaneous cyclization and a final reduction by the pyrroline-5-carboxylate reductase ProC or GriH (Fig. 1a)[4]. We previously reasoned that the genes required for synthesizing the (2 S,4 R)−4-MePro building block could be utilized to explore compounds containing such an unusual residue[4]. An earlier study discovered new natural products containing the diastereoisomer (2 S,4 S)-4-MePro from cyanobacteria employing different biosynthetic genes from the Nostopeptolide pathway as genome mining bait[5].

In this work, we address the question of whether novel (2 S,4 R)-4-MePro-containing compounds from Actinobacteria with improved bioactivity and physicochemical properties could be discovered via genome mining using the gene pair *griE* and *griF* as the probe, considering that the GM producer *Streptomyces* sp. DSM40835[6,7] belongs to the order *Actinomycetales*, the most significant industrial source of natural product drugs[8,9]. We discover the previously unknown mycoplanecin (MP) biosynthetic gene cluster (BGC) and isolate several MP derivatives that indeed exhibited significantly improved

pharmaceutical properties compared to GM. Analysis of a co-crystal structure and biophysical assays prove that MPs bind to DnaN in the nanomolar range. In addition, we comprehensively characterize in vitro three interwoven and uncommon biosynthetic pathways to the non-natural amino acids building blocks of MPs: L-homoleucine, L-homonorleucine, and (2 S,4 R)-4-ethylproline ((2 S,4 R)-4-EtPro).

## Results

### Genome mining employing *griEF* as bait identifies the mycoplanecin biosynthetic pathway

Our search for BGCs including *griE* and *griF* homologs in all actinobacterial genomic sequences in the NCBI nuccore and assembly revealed dozens of hits (Fig. 1b and Supplementary Table 1). This mining strategy was effective as it found the BGCs known to produce compounds containing 4-MePro, such as GM[4] and acyldepsipeptide (ADEP) antibiotics[10] (Fig. 1b, c). Among the unknown hits which are mainly nonribosomal peptide synthetase (NRPS) or NRPS hybrid BGCs, one hit similar to the GM pathway drew our attention because it exhibited significant differences. However, only a partial BGC could be recovered due to the low genome sequence quality. This short contig containing *griEF* homologous genes belongs to the genome of *Actinoplanes awajinensis* subsp. *mycoplanecinus* NRRL B-16712[11]. Additional genes that are absent in the GM pathway were identified, including such encoding a radical *S*-adenosylmethionine (SAM) protein, an acyl carrier protein (ACP), and a 3-ketoacyl-ACP synthase (KAS) III homolog (Fig. 1b, d). Fortuitously, we found that this NRRL strain is equivalent to *A. awajinensis* subsp. *mycoplanecinus* subsp. nov. ATCC 33919 (Strain No. 41042) producing anti-mycobacterial MP peptides discovered in the 1980s[12–16]. Notwithstanding the structural similarity

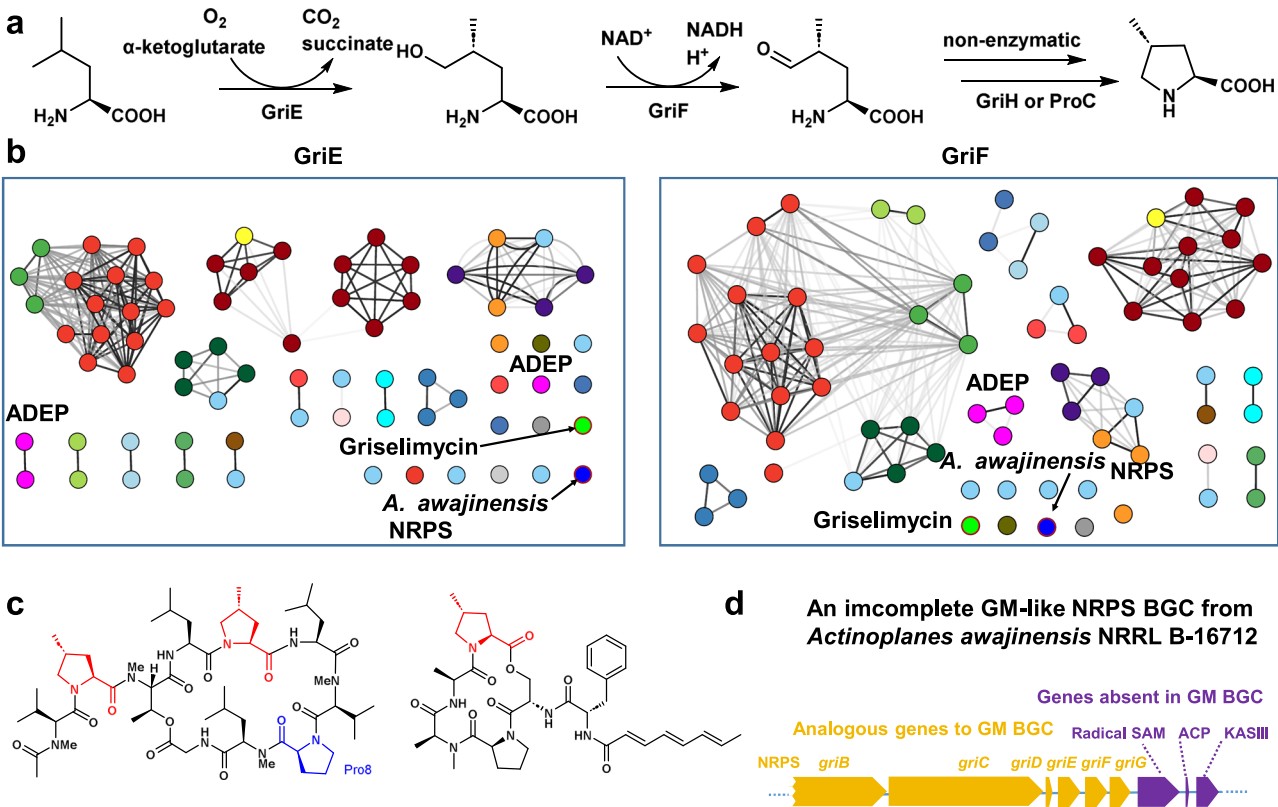

**Fig. 1 | Discovery of mycoplanecin biosynthetic gene cluster (BGC) by sequence similarity network (SSN) analysis. a** GriE and GriF are essential for biosynthesis of (2 S,4 R)-4-MePro in the griselimycin (GM) pathway. **b** Sequence similarity networks for GriE (left) and GriF (right). All edges corresponding to sequence identity below 80 % were removed. The color of the edge represents the sequence similarity, with darker color corresponding to higher similarity. Node colors correspond to the type of the biosynthetic gene cluster (Supplementary Table 1). **c** Chemical structures of GM and ADEP1 which contain (2 S,4 R)-4-MePro. **d** The incomplete NRPS BGC from *Actinoplanes awajinensis* NRRL B-16712 shows similarity to GM BGC but also contains several different genes.

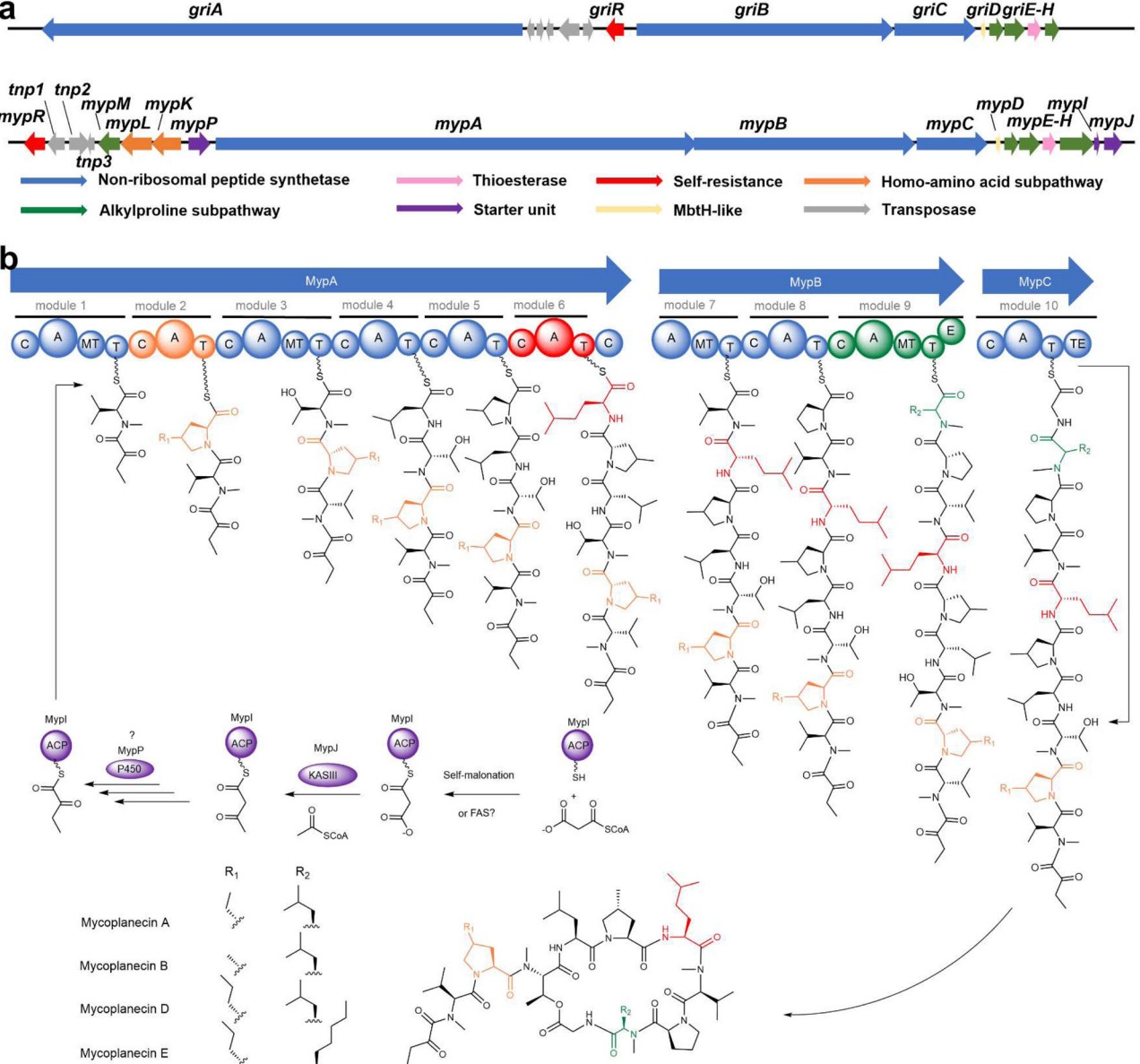

**Fig. 2 | MP biosynthetic gene cluster and proposed biosynthetic pathway. a** The gene organization and comparison of GM and MP biosynthetic gene clusters. **b** The MP biosynthetic pathway proposal. The A domains and corresponding atypical building blocks are shown in orange, red, or olive green, respectively. The proteins involved in the hypothetic biosynthetic pathway of the α-ketobutyrate precursor are in purple. For each module, the domains are: C = condensation, A = adenylation, MT = methyltransferase, T = thiolation domain, also known as peptidyl carrier protein domain in NRPS, E = epimerase, TE = thioesterase.

to GM, MPs possess unique characteristics, including the exceptional N-terminal α-keto butyric moiety, as well as the other two 4-alkylprolines: 4-EtPro and 4-propylproline (4-PrPro), homoleucine (5-Methyl-norleucine), and N-methyl-homonorleucine (N-methylheptanoic acid) (Fig. 2).

## Characterization of the MP BGC possessing a self-resistance gene

Because of the highly fragmented genome of B-16712 we PacBio sequenced ATCC33919 and obtained the intact 59 kb BGC of MP by searching for the *griEF* hit sequence in its completed genome (Fig. 2a). The putative MP BGC comprises 15 genes divided into three hypothetical operons that encode NRPSs, precursor biosynthetic enzymes, and a self-resistance protein (Fig. 2a, Supplementary Table 2). We found a DnaN-encoding gene *mypR* nearby transposase genes, which was also observed for *griR* in the GM pathway[4]. Furthermore, *mypR* is found to be the second DnaN gene in the genome of strain ATCC33919,

implying that MPs are potential DnaN inhibitors, and *mypR* confers self-resistance against MPs[3]. The majority of genes are clustered in one operon, which encodes NRPSs MypABC, the MbtH protein MypD, redox enzymes MypP (P450), MypE (hydroxylase), and MypF (dehydrogenase), the thioesterase MypG, the discrete ACP MypI, and the ketosynthase MypJ, as well as a radical SAM enzyme MypH. Inconsistencies in the assembly of the NRPS gene *mypA* caused by repetitive sequences were solved by comprehensive PCR amplification and restriction digestion analysis (Supplementary Figs. 1–3). The three gene operon comprising *mypKLM* is located upstream adjacent to the primary operon and oriented in opposite direction (Fig. 2a). MypK and MypL show sequence similarity to enzymes involved in the branched-chain amino acid (BCAA) biosynthesis. MypM is a SAM synthetase probably involved in regenerating the cofactor required by MypH.

MypA, MypB, and MypC comprise six, three, and one single NRPS module, respectively. The substrate specificity prediction of adenylation (A) domains correctly suggested the incorporation of Thr3, Leu4,

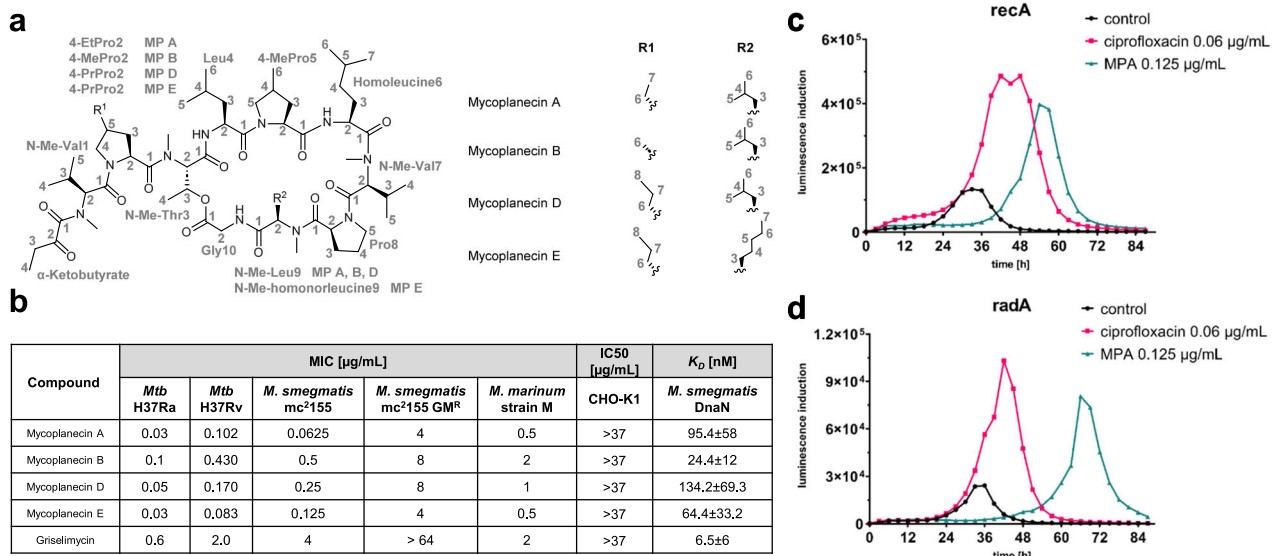

**Fig. 3 | Chemical structures and bioactivity of MPs. a** The chemical structures of MPs isolated in this study. **b** The comparison of bioactivity of MPs and GM against Mycobacteria and dissociation constants against DnaN from *M. smegmatis* mc²155. The $K_D$ values with standard deviation were calculated from three independent microscale thermophoresis measurements. **c** The strong induction of SOS response through RecA in *M. smegmatis* treated with 2x MIC MP A. **d** The strong induction of DNA repair through RadA in *M. smegmatis* treated with 2x MIC MP A.

| Compound | MIC [µg/mL] | | | | | IC50 [µg/mL] | $K_D$ [nM] |
|---|---|---|---|---|---|---|---|
| | *Mtb* H37Ra | *Mtb* H37Rv | *M. smegmatis* mc²155 | *M. smegmatis* mc²155 GM$^R$ | *M. marinum* strain M | CHO-K1 | *M. smegmatis* DnaN |
| Mycoplanecin A | 0.03 | 0.102 | 0.0625 | 4 | 0.5 | >37 | 95.4±58 |
| Mycoplanecin B | 0.1 | 0.430 | 0.5 | 8 | 2 | >37 | 24.4±12 |
| Mycoplanecin D | 0.05 | 0.170 | 0.25 | 8 | 1 | >37 | 134.2±69.3 |
| Mycoplanecin E | 0.03 | 0.083 | 0.125 | 4 | 0.5 | >37 | 64.4±33.2 |
| Griselimycin | 0.6 | 2.0 | 4 | > 64 | 2 | >37 | 6.5±6 |

Pro8, Leu9, and Gly10 (Fig. 2b). The four N-methyltransferase (MT) domains align with the N-methylation of Val1, Thr3, Val7, and Leu9 in MPs. However, modules 1 and 7 were inappropriately predicted to activate Thr instead of Val. Intriguingly, several nonproteogenic amino acid building blocks were found in the biosynthesis of MPs. A6 is predicted to activate ornithine, but ʟ-homoleucine was found incorporated in MPs in this position. The A9 domain has a promiscuous substrate tolerance since ʟ-N-methyl-homonorleucine and ʟ-N-Methyl-Leu are present in different MPs at this position. The substrate specificity of A2, A5, and A8 was predicted as proline, indicating that the current bioinformatic analysis could not discriminate proline from different alkyl-prolines[4]. The substrate specificity of A2 is flexible as three different 4-alkylproline residues are incorporated in different MP derivatives. However, it is also possible that different 4-alkylproline residues are formed by post-modifications of 4-MePro in MP B (Fig. 2b). It is noteworthy that uncommon genes responsible for lipopeptide chain initiation are found in the MP pathway, partially similar to the proposed pathway for calcium-dependent antibiotics (CDAs)[17]. Here, it is hypothesized that the holo-MypI (ACP) is either malonylated by the malonyl-CoA-acyl carrier protein transacylase (MCAT) FabD or functions as a self-priming ACP that widely exists in fatty acid biosynthesis[18,19]. Subsequently, the KAS III-like enzyme MypJ extends the acyl chain by a molecule of acetyl-CoA to form the acetoacetyl-*S*-ACP species, which is probably reduced to butyryl-*S*-MypI by reductases from the type II fatty acid biosynthesis. A P450-dependent oxidase encoding gene *mypP* is assumed to be responsible for the cryptic α-keto group formation, albeit the mechanism and oxidation timing are still unexplored (Fig. 2b, Supplementary discussion).

### Isolation and structure elucidation of MPs with unusual building blocks

Mycoplanecins were first investigated four decades ago due to their activity against Mycobacteria while possessing low cytotoxicity, but no subsequent studies are known from the literature. The absolute configurations of MPs are not completely clear[12–16], nor is the biosynthesis or the molecular target. Given the presence of the putative self-resistance gene *mypR* and the finding of novel biosynthetic genes, we re-isolated these potent anti-tubercular compounds aiming to identify unprecedented derivatives for subsequent characterization.

The three most abundant derivatives were purified and structurally elucidated as the previously described MP A, B, and D (Fig. 3a)[12,13,15]. A complete set of 1D and 2D nuclear magnetic resonance (NMR) experiments was acquired for each derivative to underpin the structures and to supplement previous MP spectral data (Supplementary NMR Data)[12,13,15]. The last derivative isolated, MP E, is structurally novel with a 4-PrPro and a N-methyl-homonorleucine residue incorporated, the largest of the MPs (Fig. 3a). Comprehensive studies of COSY, HSQC and HMBC spectra indicate the presence of α-ketobutyrate, N-methyl-valine, leucine, 4-methylproline, 5-methylnorleucine, proline, glycine and N-methyl-threonine analog to the known derivatives MP A, B and D. Furthermore, the spectral data reveal that MP E contains the same 4-propylproline moiety as described in MP D, which is substituted by 4-methylproline in MP A and 4-ethylproline in MP B. N-methylleucine found in MP A, B and D is replaced by the rare N-methylheptanoic acid building block in MP E. This moiety is characterized by the three methylene signals at δ(¹H) = 1.16 (2H, m) δ(¹³C) = 38.9, δ(¹H) = 1.26 (2H, m) δ(¹³C) = 27.3 and δ(¹H) = 1.64 (2H, m) δ(¹³C) = 24.9 as well as the more shielded methyl signal at δ(¹H) = 0.90 (3H, m) δ(¹³C) = 14.1 ppm. Besides this, the chemical shifts of the remaining structure of MP D and MP E exhibit a deviation of less than 0.12 ppm for δ(¹H) and 2.5 ppm for δ(¹³C) (Supplementary NMR Data). We proposed the stereochemistry of all derivatives through Marfey's analysis and biosynthesis analysis, confirming that MP A comprises N-methyl-valine, 4-EtPro, leucine, 4-MePro, proline, N-methyl-threonine in ʟ-configuration while N-methyl-leucine is the only ᴅ-configured amino acid underlined by the only epimerization (E) domain (Fig. 2b and Supplementary Figs 21-28, 35–41, 48-55, 60-67).

### MPs show low nanomolar anti-*Mtb* potency

Importantly, bioactivity testing of the MPs against the attenuated *M. tuberculosis* strain H37Ra uncovered superior activity in the nanomolar range (minimum inhibitory concentration (MIC) from 30 to 100 ng/mL (Fig. 3b)). Additional MPs tested against *M. smegmatis* revealed potent activity from 0.0625 to 0.5 µg/mL. Although MPs remain relatively effective against a GM-resistant *M. smegmatis* strain[3] with MIC values ranging between 4 to 8 µg/mL, there was a clear MIC shift (ranging between 16 to 64-fold), indicating that mycoplanecins have the same target as GM. It should be noted that resistance against griselimycins is

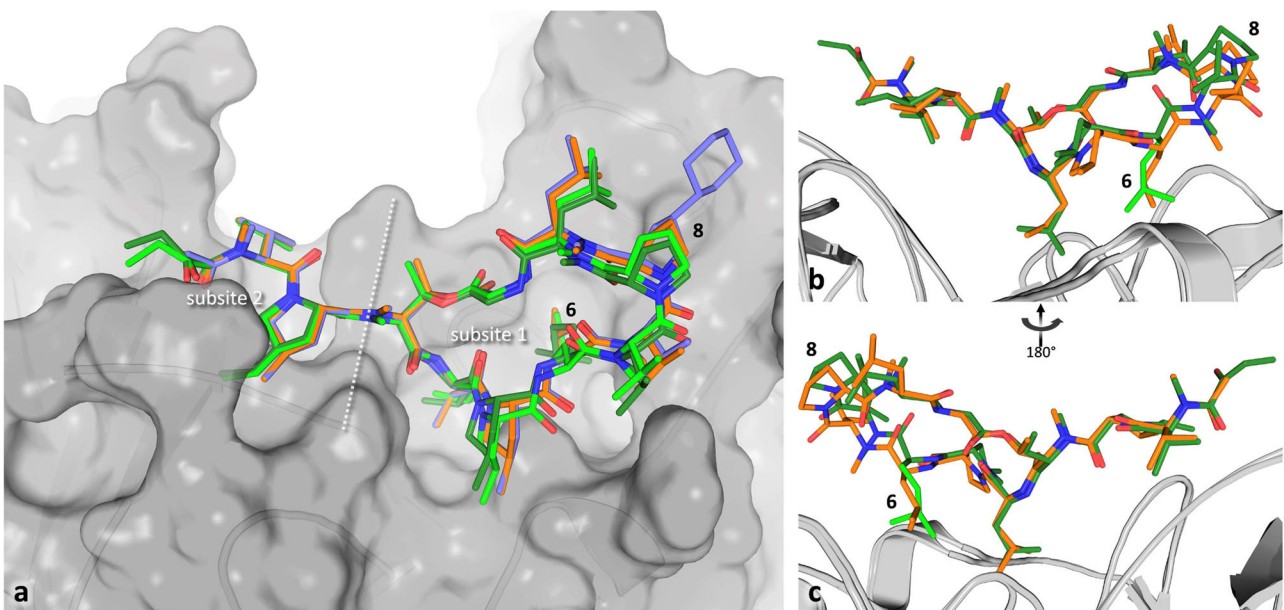

**Fig. 4 | Binding of MP A to ecDnaN. a** Superimposition of the ecDnaN co-crystal structures containing MP A (8CIZ, dark/light green for the two entities in the asymmetric unit), GM (8CIX, orange), and CGM (8CIY, violet). The protein surface is shown in gray. The two subsites of the binding pocket are separated by a dashed white line. The structures were superimposed on the level of single amino acid chains of DnaN. **b** Binding comparison of GM (orange) and MP A (green) to ecDnaN (gray cartoon). The 5-Methyl-norleucine residue at position 6, that pushes MPA away from the protein in comparison with leucin 6 in GM is highlighted in light green. **c** View of the GM/MP A binding comparison turned 180° around the Y axis.

associated with severe fitness loss in the mutants[3]. While MPs were less potent against the virulent *M. tuberculosis* strain H37Rv than the attenuated H37Ra, they still exhibit an impressive ~24-fold more potent bioactivity than GM (Fig. 3b). Surprisingly, the best MIC was 0.083 µg/mL for the new congener MP E, while the MIC of GM is 2 µg/mL. MPs were additionally tested against a panel of Gram-positive (*S. aureus* Newman and *B. subtilis* DSM 10) and Gram-negative (*E. coli* BW25113, *C. freundii* DSM 30039 and *A. baumannii* DSM 30008), but exhibited no activity up to 64 ug/mL (Supplementary Table 6). The mycobacteria-specific inhibitory bioactivity of MPs highlighted the high selectivity of this compound class, also concerning members of the human microbiota that are likely spared in the potential use of MP in TB therapy. In addition, MPs showed no cytotoxicity against CHO-K1 cells up to a concentration of 37 µg/mL, which underpins the high potential of MPs in the application to combat mycobacterial infections. DNA damage response after exposure to MP A was evaluated in *M. smegmatis* reporter strains harboring mycobacterial promoters that drive the expression of DNA damage-inducible genes, *recA* and *radA*. Strong induction of luminescence was observed for the reporter strains upon treatment with 2x MIC MP A, indicating that the target of MPs is involved in DNA metabolism that results in genotoxic stress. (Fig. 3c, d).

**MPs bind to DnaN in nanomolar affinity**
The homology of the self-resistance gene *griR* from the GM BGC with *mypR* in the MP BGC and MPs' structural analogy to GM implied that DnaN is the molecular target of MPs. We determined the binding affinity of all isolated MPs as well as GM to DnaN by microscale thermophoresis (MST). The highest affinity to *M. smegmatis* DnaN (msDnaN) was observed for GM with $K_D = 6.5 \pm 5.9$ nM. Lower DnaN binding affinities were determined for MPs as the $K_D$-value for MP A was $95.4 \pm 58.0$ nM while MP B showed strongest binding to msDnaN among the congeners with a $K_D = 24.4 \pm 11.9$ nM. The other two MPs showed a similar binding range to msDnaN (Fig. 3b, Supplementary Fig. 4). Interestingly, GM manifests a slightly stronger DnaN binding affinity than the MPs but is less active than MPs against mycobacteria. This intriguing phenomenon is unresolved but is likely a result of MPs'

improved physicochemical properties including increased lipophilicity, which is important for TB activity.

**Crystal structure of DnaN in complex with MP A**
Although the co-crystal of msDnaN and MP A could not be obtained after intensive attempts, DnaN of *E. coli* (ecDnaN) was overproduced in *E. coli* and co-crystallized with MP A, GM and cyclohexyl-GM (CGM). Despite sharing only 29 % sequence identity with msDnaN, the binding pocket of ecDNA shows a sufficient degree of conservation (Supplementary Fig. 5). The crystal structures were solved at resolutions of 2.12 Å (MP A), 1.65 Å (GM) and 1.50 Å (CGM) (Supplementary Fig. 6 and Supplementary Tables 8, 9). MP A was found to bind in the same mode to ecDnaN as GM or CGM. The binding pocket can be divided into a larger (subsite 1) and a smaller cavity (subsite 2) and as has been shown previously for GM and mycobacterial DnaN, all compounds occupy both subsites of the binding pocket. All three substances form hydrogen bonds to only 2 – 3 protein residues, but the additional oxygen in the N-terminal α-keto butyrate residue of MP A provides a second hydrogen bond acceptor for Arg365 (Supplementary Fig. 7), which is also conserved in mycobacterial DnaN. The compounds bind mainly via van der Waals interactions with interfaces ranging from 550 Å² for GM to 590 Å² for CGM to 600 Å² for MP A. Compared to available structures of uncomplexed DnaN, only a few residues undergo significant conformational changes upon binding of the compounds - most notably Met362, which has been described to function as a gate between the subsites (Supplementary Fig. 8)[20]. The linear, N-terminal part of the ligands in subsite 2 superimposes almost perfectly, while MP A seems to be tilted slightly (approximately 7-10 ° compared to GM/CGM) away from the protein in the region around Pro8. This tilting can be attributed to the longer 5-methyl-l-norleucine residue of MP A at position 6, whereas leucine is present at the same position in GM and CGM (Fig. 4 and Supplementary Fig. 9). The comparison with the previously published mycobacterial GM/DnaN complexes shows that whereas the N-terminal parts of the ligands in subsite 2 superimpose very well, there is higher variance on the opposite side of the molecules around Pro8 (Supplementary Fig. 9). In the complex with msDnaN, GM

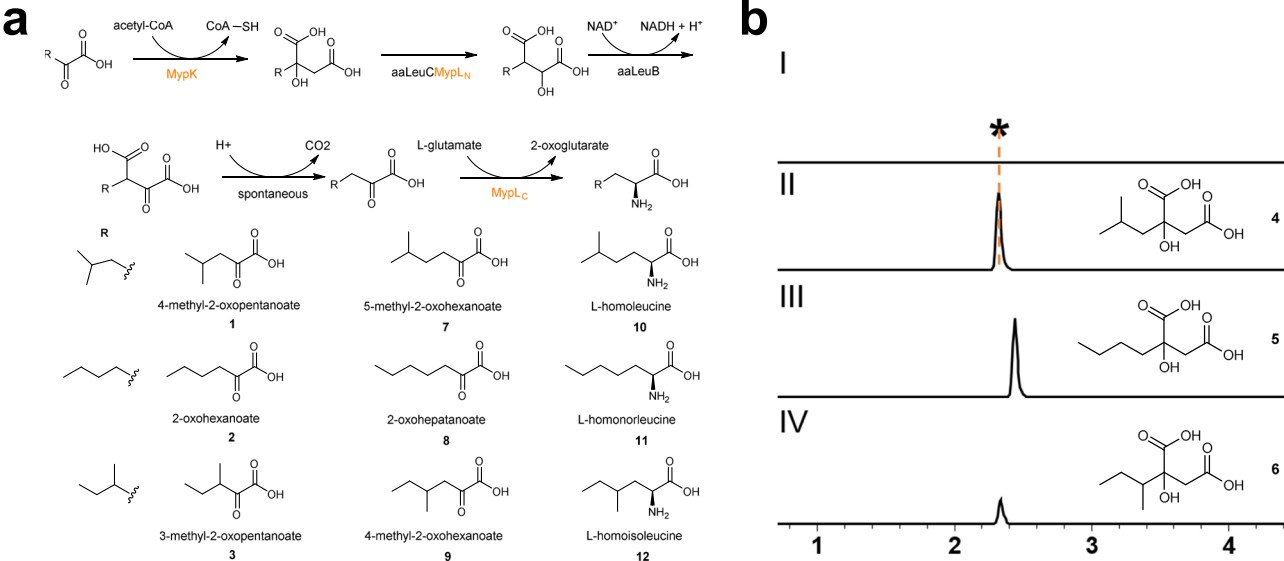

**Fig. 5 | The homo-amino acid building blocks of MPs are built by a pathway mimicking BCAA biosynthesis. a** Proposed subpathway and intermediates of L-homoleucine (**10**), L-homonorleucine (**11**) and L-homoisoleucine (**12**) formation. The names of proteins from the MP pathway involved in the homo-amino acid sub-pathway are highlighted in orange. **b** In vitro assays of the purified protein MypK.

Extracted ion chromatograms (EICs) corresponding to seven-carbon α-keto acids detected in negative ion mode ([M-H]⁻, m/z 189.07). No formation of 2-isobutylmalate (**4**) could be detected with boiled MypK and substrate **1** as control (I). The formation of **4**, 2-n-butylmalate (**5**), and small amount 2-sec-butylmalate (**6**) were observed upon incubation of MypK and **1** (II), **2** (III), and **3** (IV), respectively.

adopts a conformation very similar to that of MP A in the ecDnaN complex, although GM lacks the 5-methyl-l-norleucine at position 6 and the residues lining the protein pocket around this position are conserved between mycobacteria and *E. coli*. In complex with *M. tuberculosis* DnaN, the tilt of GM away from the protein is less pronounced, instead the macrocycle seems to be tilted approximately 7° towards positon 7 in comparison to the msDnaN complex. Although differences in crystal packing make a direct comparison of the binding strength difficult, the linear part of the ligands (1-3) and the residues adjacent to the bifurcation (4-5, 10) seem to be bound tighter/show less flexibility than the opposing residues (6-9) in all structures of DnaN/GM complexes (Supplementary Fig. 10).

## In vitro reconstitution of the α-keto acid elongation subpathway

NRPS megasynthetases exhibit the unique feature of incorporating unusual and nonproteinogenic amino acids into bioactive peptidic molecules. For instance, 4-ethyl-L-proline/4-propyl-L-proline, L-homoleucine, and L-homonorleucine as present in MPs, often contribute to the bioactivities whereas their biogenesis remains mysterious. We proposed the biosynthesis of the homo-amino acid residues in MPs as biosynthetically sharing an analogous process to leucine biosynthesis, as previously also suggested for a few other homo-amino acids. Two homologous genes of leucine biosynthesis were identified within the MP BGC. *mypK* encodes a 2-isopropylmalate synthase-like enzyme that likely catalyzes the condensation of different six-carbon α-keto acids, including 4-methyl-2-oxopentanoate (**1**), 2-oxohexanoate (**2**), and 3-methyl-2-oxopentanoate (**3**) with acetyl-CoA. The other gene encodes a didomain enzyme MypL, which contains a putative N-terminal isopropylmalate isomerase subunit (LeuD) homolog and a C-terminal pyridoxal phosphate (PLP)-dependent transaminase domain. Although the dedicated dehydrogenase (*leuB*) and the other isopropylmalate isomerase subunit (*leuC*) genes were not found in the MP pathway, the broad substrate tolerance of leucine biosynthesis enzymes was nevertheless considered to be important for homo-amino acids biosynthesis as well[21]. Thus, MypK, MypL, together with aaLeuB and aaLeuC from primary metabolism of *A. awajinensis* ATCC 33919, were proposed to biosynthesize the homo-amino acid residues (**10, 11**, and **12**) (Fig. 5a).

We set out to study these intriguing steps in vitro because *Actinoplanes* species are notoriously recalcitrant to genetic manipulation, including strain ATCC33919. The acceptance of **1, 2**, and **3** as substrates by the recombinant His-tagged MypK was demonstrated by the newly detected mass peaks showing 60 Da mass increase, corresponding to the compounds 2-isobutylmalate (**4**), 2-n-butylmalate (**5**), and 2-sec-butylmalate (**6**), respectively (Fig. 5b). Moreover, MypK exhibits a preference for **1** compared to **2** and **3**, which was verified by measuring enzyme kinetics (Supplementary Table 10). A new peak with m/z shift of 1 Da corresponds to the expected mass of L-homoisoleucine (**12**) and was detected by LC-MS in the reaction of the recombinant amino-transferase MypL_C in the presence of L-glutamate as the amino group donor and 4-methyl-2-oxohexanoate (**9**) as the substrate. MypL_C was tested under the same condition revealing an extensive range of pro-teinogenic amino acids as amino donors (Supplementary Fig. 11)[22].

The substrate acceptance of 4-methyl-2-oxohexanoate for MypL_C indicated that MypL_C was likely involved in forming **10, 11**, and **12**. The successful purification of the remaining enzymes enabled the total in vitro reconstitution of homo-amino acid biosynthesis in one pot (see below and Supplementary Fig. 12). MypL_N was co-purified with aaLeuC in an intensive brown color, indicating the binding of an iron-sulfur cluster. The one-pot reaction was executed in three stages initiated by MypK, followed by the reaction of aaLeuC and MypL_N co-purified protein, and aaLeuB together with β-Nicotinamide adenine dinucleo-tide in its oxidized form (NAD⁺). After the last stage reaction with MypL_C, D-FDLA was used to derivatize potential amino acid products. The reaction product using **1** as the substrate had the same retention time compared with a peak with the same mass from the hydrolyzed and D-FDLA derivatized MP D, suggesting the formation of **10**. Similarly, the mass corresponding to the derivatized **11** or **12** was detected from the reaction using **2** or **3** as substrate, respectively. (Fig. 5b and see below).

## In vitro reconstitution of the 4-alkylproline subpathway

Previously, 4-EtPro and 4-PrPro were characterized to be derived from L-tyrosine in the biosynthesis of lincomycins (Supplementary Figs. 13 and 14a)[23], however, no homologous genes were found in the MP pathway. As reported in GM biosynthesis, (2 S,4 R)−4-MePro was the

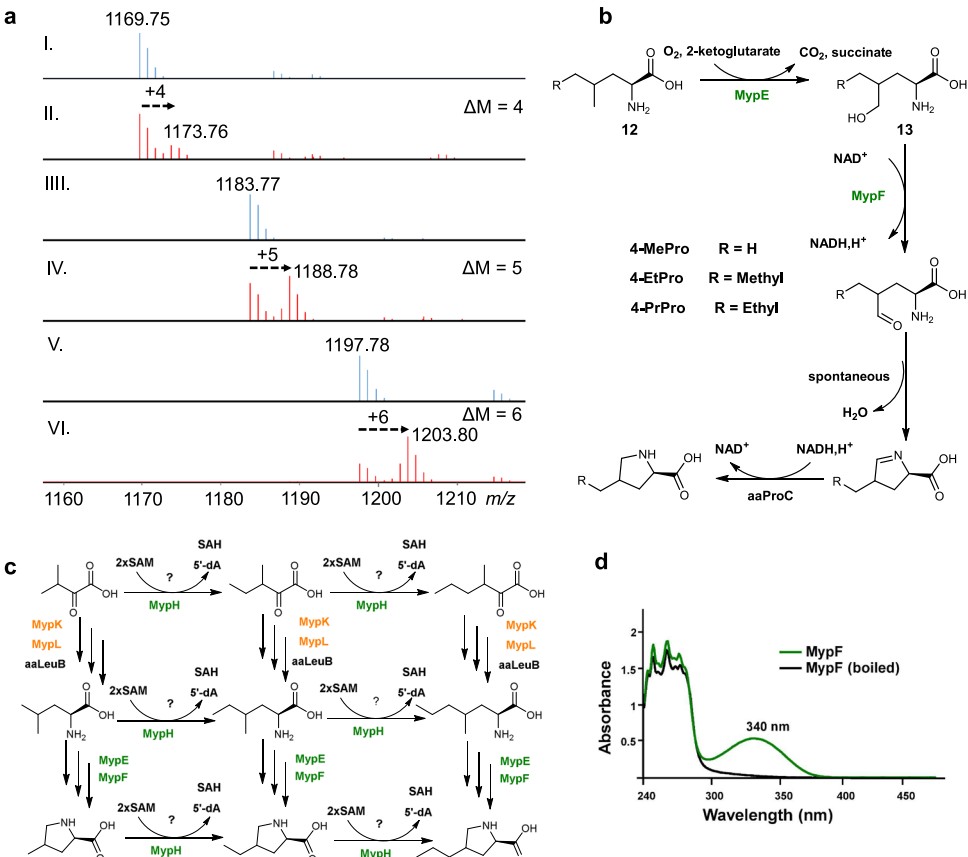

**Fig. 6 | The biosynthesis of alkylproline residues in MPs. a** MS analysis of production of MPs by fermentation without (I: MP B, III: MP A, V: MP D) or with (II: MP B, IV: MP A, VI: MP D) L-Methionine-(*methyl*-[13]C). **b** Proposed subpathway of 4-EtPro and 4-PrPro formation in MP biosynthesis. **c** Potential methylation timepoints by MypH. **d** UV/Vis spectra of MypF and heat-inactivated MypF in the assay.

product of a series of enzymatic modifications of leucine (Fig. 1a)[4]. A pair of genes *mypE* and *mypF* from the MP pathway shows homology with the genes involved in the 4-MePro biosynthesis, including *ldoA* and *nosE* from Cyanobacteria, and *griE* and *griF* in GM biosynthesis[4,24]. However, the biosynthesis pathways of 4-EtPro and 4-PrPro in MPs remained elusive.

By supplementing the culture of strain ATCC33919 with L-Methionine-(*methyl*-[13]C) and subsequent MS analysis of the resulting mycoplanecins, a 1 Da difference between control and isotope-labeled MP B ( + 4 Da) A ( + 5 Da) and D ( + 6 Da) suggested the extra carbons of 4-EtPro and 4-PrPro are derived from methionine that is used for radical-SAM based methylation of 4-MePro (Fig. 6a). We speculated that MypE/F form 4-MePro as described in the GM pathway (Fig. 6b). However, the timing of subsequent methylation by the radical SAM MT MypH is still elusive (Fig. 6c). MypH might iteratively methylate the unreactive methyl-carbon of nascent substrates at different stages, including the α-keto acids before chain elongation, the homo-amino acids before hydroxylation by MypE, or 4-MePro (Fig. 6c). In addition, we can not exclude that 4-EtPro and 4-PrPro-containing MPs could be produced by modifying MP B by MypH as a post-assembly tailoring enzyme.

The formation of 4-hydroxymethyl-L-homoisoleucine (**13**) was observed as a new mass with a 16 Da increase in the D-FDLA derivatized reaction product of recombinant MypE incubated with homoisoleucine (stereoisomer mixture) since the pure (2*S*,4*R*)-homoisoleucine is not commercially available. Surprisingly, three individual new peaks with the same *m/z* were observed after derivatization by D-FDLA, suggesting loose stereospecificity of MypE (Fig. 7b). A series of structural analogs were investigated for substrate tolerance of MypE,

but no conversion was detected with L-valine, D-isoleucine or D-allo-isoleucine. In contrast, a trace amount of products of D-leucine and L-isoleucine, and nearly complete conversion of L-leucine and L-allo-isoleucine was observed by LC-MS (Fig. 7b). These results indicated MypE is a promiscuous enzyme recognizing substrate stereoisomers, however, preferring L-forms of amino acids, especially the L-allo form amino acid in the case of two stereocenters.

The reaction of MypF proceeded well with the substrate 4-hydroxymethylhomoisoleucine (**13**) from *E. coli* overexpressing recombinant MypE and incubated with homoisoleucine, as determined by a significant rise of absorbance at 340 nm, indicating NADH formation (Fig. 6d). A new signal corresponding to the D-FDLA-derivatized (2*S*,4*R*)−4-EtPro was observed in the coupled assay with MypE, MypF, and aaProC (pyrroline-5-carboxylate reductase from the MP producer), which is consistent with the mass and retention time of the derivatized hydrolysis product of MP D (Fig. 7a).

## Discussion

Given the promising activity of griselimycin as a potential anti-TB drug, its innovative target DnaN, and the involvement of nonproteogenic amino acids in target binding, we became interested in identifying additional natural products exhibiting similar chemical moieties. Investigation of all publically available actinobacterial genomes using (2*S*,4*R*)−4-MePro biosynthesis genes[4] unveiled a previously unknown BGC with additional interesting precursor biosynthetic genes compared to the GM pathway. This finding inadvertently disclosed the mycoplanecin BGC and correlated it to these compounds initially discovered in the 1980s with potent and specific anti-tuberculosis bioactivities[12–16].

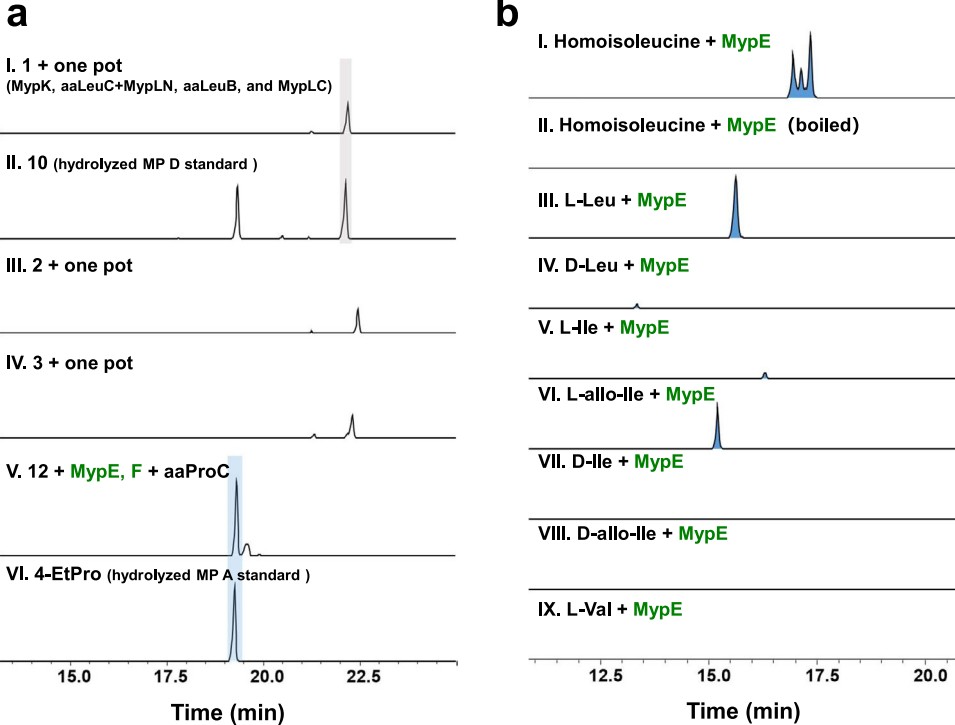

**Fig. 7 | In vitro reconstitution of the biosynthesis of homo-amino acid and alkylproline residues. a** In vitro analysis of the extender unit reconstitution. EICs of both D-FDLA derivatized reaction samples and hydrolyzed MP D and A as control were analyzed. Derivatized **10** (*m/z* 440.213) was detected in one pot incubation (I) with **1**, MypK, aaLeuC+MypL$_N$, aaLeuB and MypL$_C$, and from hydrolyzed and derivatized MP D standard (II). The peak at ~19.5 min (retention time) in II corresponds to Marfey's derivatization product of N-methyl-D-leucine. Derivatized **11** and **12** products (*m/z* 440.213) were found using **2** and **3** as substrate in one pot assay, respectively (III, IV). The mass (*m/z* 438.198) corresponding to 4-EtPro formation was observed in reaction of **12** with MypE, MypF and aaProC (V), and from

hydrolyzed MP A standard (VI). **b** In vitro analysis of MypE (EICs). The 4-hydroxymethylhomoisoleucine (**13**) (*m/z* 456.208) was detected after incubation of MypE and homoisoleucine (I), but not with boiled MypE (II). The formation of 4-hydroxy-L-Leu (*m/z* 442.193) after incubation of MypE and L-Leu (III). Trace amount of 4-hydroxy-D-Leu (*m/z* 442.193) after incubation of MypE and D-Leu (IV). Trace amount of 3-hydroxy-L-allo-Ile (*m/z* 442.193) after incubation of MypE and L-Ile (V). The formation of 3-hydroxy-L-allo-Ile (*m/z* 442.193) after incubation of MypE and L-all-Ile (VI). No product was detected with using D-Ile, D-allo-Ile and L-Val as substrate (VII-IX).

Here, four MPs, including one new congener with the most potent anti-*Mtb* bioactivity, were isolated and found to contain several unusual nonproteogenic amino acid residues, including three different 4-alkylprolines. Marfey's analysis plus the NRPS E domain for the first time proved the absolute stereochemistry of the MPs. These amino acid residues, especially (2*S*,4*R*)-4-EtPro and (2*S*,4*R*)-4-PrPro, are scarce in natural products. Interestingly, the 4-PrPro and 4-EtPro in lincomycins instead derive from L-tyrosine degradation as catalyzed by a subcluster of six genes[25] which are not present in the MP BGC.

A step-by-step in vitro total reconstitution of the BCAA-like pathway of MP biosynthesis revealed a set of promiscuous enzymes which build unusual homo-amino acids from different 2-oxo acids. Subsequently, we in vitro reconstituted 4-EtPro using a pair of GriEF-like enzymes from the MP pathway. A feeding study with L-Methionine-(*methyl*-$^{13}$C) suggested the extra carbons of 4-EtPro and 4-PrPro over 4-MePro in MPs likely to come from radical SAM methylation by MypH. The successful total in vitro reconstitution of 4-EtPro formation starting from **3** implied a potential 4-alkylproline pathway with a yet uncertain methylation timepoint (Figs. 6c and 7a).

Considering the substrate tolerance of all related Myp enzymes as shown for homo-amino acids and 4-EtPro reconstitution, one reasonable pathway is MypE hydroxylation of **12** to form **13** and subsequent oxidation and spontaneous cyclization yielding the pyrroline intermediate that is further reduced to (2*S*,4*R*)-4-EtPro. The 4-PrPro residue can be produced following the same logic. However, it requires the previously unknown 3-methyl-2-oxohexanoic acid as the starting substrate, which needs experimental proof in the future (Fig. 6c). It is still mysterious when exactly MypH performs its function since potentially

the radical methylation can either take place on the 4-alkylproline precursor or as a tailoring step to modify MP B directly.

Intriguingly, MPs showed significantly more potent anti-*Mtb* bioactivity than GMs. We demonstrated that the new congener MP E shows the most potent MIC (0.083 μg/mL), exhibiting 24-fold higher potency against *Mtb* H37Rv than GM. We showed MP A in our hands has a four-fold lower MIC (0.102 μg/mL) against *Mtb* H37Rv than initially reported (0.39 μg/mL)[14]. The analogous chemical structures of MPs and GMs align with their identical molecular target site within DnaN. In comparison to GM and CGM, the higher potency of MP A might be attributed to the increased hydrophobicity of the substance and the larger van der Waals interface with DnaN. The 5-methyl-L-norleucine at position 6 might also play a role by pushing the generally less-tight bound region around Pro8 out of the binding pocket but stabilizing the tighter interaction around the ligand's bifurcation. The co-crystal structure of ecDnaN bound with MP A and the nanomolar binding affinity of MPs to msDnaN validates DnaN as the molecular target. The binding mode is similar to GM and CGM, albeit MPs showed significantly stronger anti-Mtb bioactivity. This finding is of great importance as GM development towards a TB drug was stopped due to unwanted but target-independent side effects (unpublished results). As mycoplanecins show significantly lower MICs they might be useful to overcome this limitation as the potential therapeutic window over the side effect is significantly improved.

TB is still one of the leading public health threats, even during the global COVID-19 pandemic. Drug-resistant strains of *Mtb* further deepen the crisis because we are running out of our antibiotic arsenal. The low cytotoxicity and specific anti-mycobacterial activity

thus make MPs promising candidates to be developed as potent and safe anti-TB drugs that are likely not detrimental to human microbiota. The discovery of highly potent antibiotics exhibiting the rare target DnaN, and the unusual biosynthesis of the molecular building blocks sets the stage for continuous efforts into future mycoplanecin development.

## Methods

### Sequence similarity network analysis

Sequence similarity networks were constructed for all homologs of GriE and GriF found in biosynthetic gene clusters (BGCs) in all species of the phylum Actinobacteria. To this end, we conducted a BLAST (v. 2.9.0)[26] search with a threshold e-value of $10^{-5}$ and filtered for hits where the similarity is found for at least 75% of the sequence of the corresponding protein. We have retrieved the corresponding genomes or genome fragments and predicted BGCs with antiSMASH (v. 5.1.2)[27]. Only GriE/GriF homologs within BGCs were retained. The protein sequences were aligned with MAFFT (v. 7.450)[28] and pairwise sequence similarities were calculated. The sequence similarity networks were visualized with Cytoscape (v. 3.9.1)[29].

### Strains, plasmids and reagents

The mycoplanecins producer strain *Actinoplanes awajinensis* ATCC 33919 was purchased from the American Type Culture Collection. The plasmid pET28b was purchased from Novagen. The plasmid pColdI was purchased from TAKARA. CoA esters and amino acids were purchased from Sigma-Aldrich and Enamine. The 4-ethylproline was synthesized and supplied by Sanofi. Reagents used for cultivation, expression, catalyzation, derivatization and all enzymes required for molecular cloning were purchased from ThermoFisher Scientific, NEB, Sigma-Aldrich or Chemspace.

### Strain cultivation, compound extraction and purification

Cultivation of *Actinoplanes awajinensis* was carried out in 5 ×1-L flasks using 300 mL culture volume in APL medium (1% soluble starch, 0.2% yeast extract, 1% glucose (dextrose), 1% glycerol, 0.25% corn steep liquor, 0.2% peptone (meat), 0.1% NaCl, 0.3% CaCO₃, adjusted to pH 7.2). Subsequent supercritical fluid extraction (SFE) was carried out following the protocol described by Bader et al.[30] using a Waters MV-10 ASFE system and yielding the mycoplanecins in the ethyl acetate phase. The resulting SF extract was used to purify MP A, B, D, and E using a Dionex Ultimate 3000 SL system comprising a SWPS 3000 SL autosampler, P680 pump, TCC100 column oven, PDA100 UV- detector and AFL 3000 fraction collector in combination with a XBridge Peptide BEH C18 OBD prep column (130 Å, 5 μm, 10 mm × 250 mm). The flow-rate was set to 5 mL/min with a mobile phase containing (A) 0.1% formic acid in ddH₂O and (B) 0.1% formic acid in acetonitrile heated to 45 °C. After an equilibration phase for 2 min at 5% B in A, a 30 min gradient from 50 to 80% B was adapted for separation of the MP derivatives. The column was flushed with 95% B for 2 min before subsequent re-equilibration at 5% B for 5 min.

### NMR analysis for planar structure elucidation

NMR measurements of MP A, B, D, and E were recorded using a Bruker UltraShield 500 or a Bruker Ascend 700 spectrometer equipped with a 5 mm TCI cryoprobe (¹H at 500 or 700 MHz, ¹³C at 125 or 175 MHz, respectively). ¹H, ¹³C, COSY, TOCSY, HSQC and HMBC experiments were carried out in deuterated chloroform. The NMR Spectra were recorded by Bruker TopSpin 4 and processed by ACD/Labs 2021.2.0.

### Marfey's analysis for stereochemical assignment

Marfey's derivatization of MP A, B, D, and E was performed to assign the amino acids' stereochemistry by comparing the retention time of the hydrolyzed and derivatized amino acids with respective standards. Hydrolysis of the amide bonds and subsequent derivatization of the fragments was carried out as follows: 0.1 mg of the respective MP derivative was hydrolyzed with 100 μL of 6 N HCl in a closed glass vial for 45 min at 110 °C under nitrogen. The solvent was dried at 110 °C and the residue was dissolved in 110 μL H₂O. The solution was split into two 50 μL aliquots in PP tubes and both were treated with 20 μL of 1 N NaHCO₃ followed by 20 μL of 1% 1-fluroro-2,4-dinitrophenyl-5-leucine-amide (D-FDLA and L-FDLA in acetone). After incubation for 2 h at 40 °C and 700 rpm, the reaction was stopped by addition of 10 μL of 2 N HCl and 300 μL of acetonitrile. Both samples were centrifuged for 10 min at 15 °C and $21,500 \times g$, and the supernatant was measured by UHPLC-HRMS. Analysis was carried out on a Dionex Ultimate 3000 SL system coupled to a Bruker Daltonics maXis 4 G UHR-TOF applying electrospray ionization (ESI). Separation was performed using a Waters Acquity BEH C18 column (100 × 2.1 mm, 1.7 μm) with a mobile phase consisting of (A) 0.1 % formic acid in ddH₂O and (B) 0.1 % formic acid in acetonitrile with a flow rate of 0.6 mL/min at 45 °C. A multi-step gradient was applied to separate the derivatized amino acids: 5-10 % B in 1 min, 10-35 % B in 14 min, 35–55% in 7 min, 55–80% in 3 min, hold at 80% for 1 min and re-equilibration at 5 % B for 5 min. Before entering the mass spectrometer, the flow was split to 75 μL/min. UV data were obtained at 340 nm simultaneously with the MS detection in centroid mode ranging from 250 to 3000 m/z.

### Standard LC-MS methods

UHPLC-HRMS measurements for determination of exact mass, sum formula and retention time were carried out with a Dionex Ultimate 3000 SL system consisting of SWPS 3000 SL autosampler, P680 pump module, TCC100 column oven, and PDA100 UV-detector coupled to a maXis 4 G UHR-TOF platform. The ionization mode was ESI with the following MS settings: capillary voltage 4000 V, end plate off-set -500 V, nebulizer gas pressure 1 bar, dry gas flow rate 5 L/min, dry gas temperature 200 °C, mass scan range 150–2500 m/z. The mass spectrometer was externally calibrated to sodium formate cluster using subsequent lock mass calibration. Metabolite profiling was performed using Waters Acquity BEH C18 columns (50 × 2.1 mm, 1.7 μm and 100 × 2.1 mm, 1.7 μm) connected to a Waters VanGuard BEH C18 pre-column. The standard methods for both columns are linear gradients with 5–95% ACN + 0.1 % FA over 6, 9, or 18 min. The LC-MS results were analyzed by Bruker Compass DataAnalysis V4.4.

### MIC determination

To determine the inhibitory activity of the mycoplanecins, Minimum Inhibitory concentration (MIC) was carried out. *M. tuberculosis* (*Mtb*) strain H37Ra and H37Rv, *M. marinum* (*Mm*) strain M (ATCC BAA-535) and *M. smegmatis* (*Msmeg*) strain mc²155 were cultured in 7H9 complete medium (BD Difco; Becton Dickinson) supplemented with oleic acid-albumin dextrose-catalase (OADC, 10%; BD), 0.4% glycerol, and 0.05% Tween80 at 37 °C for *Mtb* and *Msmeg* and at 30 °C for *Mm*. At mid-log phase (OD₆₀₀ between 0.4 and 0.8), cultures were harvested and centrifuged ($3700 \times g$, 10 min). For *Mtb* and *Mm*, bacterial cells were then thoroughly resuspended in 7H9 medium (10% OADC) in the absence of glycerol and Tween80 by use of a syringe and a 26-gauge syringe needle. *Mtb* and *Mm* single-cell PBS stocks were thawed and used to prepare a final suspension of 10⁵ CFU/mL. Compounds were tested in twofold serial dilutions in final 1% DMSO. *Mtb* MIC values were determined by addition of 50 μL of Alamar Blue (Promega) with 5% v/v Tween80 to each well on the sixth day of incubation followed by 16–24 h of incubation at 37 °C. Fluorescence was measured (excitation at 570 nm and emission at 590 nm). *Mtb* MIC was defined as the lowest concentration affecting an 80% reduction in fluorescence relative to the signal for the control without antibiotic. For *Mm* and *Msmeg*, MIC was determined visually as the lowest concentration where no visible growth was observed.

## Cytotoxicity evaluation

CHO-K1 cells (Chinese hamster (*Cricetulus griseus*) ACC 110) were cultured under conditions recommended by the depositor. Briefly, cells were seeded at $6 \times 10^3$ cells per well of 96-well plates in 180 µl complete medium (StableCell™ Ham's F-12, Sigma-Aldrich supplemented with 10% FBS). Each compound was tested in a serial dilution as well as the internal solvent control. After 5 d incubation, 20 µl of 5 mg/ml MTT (thiazolyl blue tetrazolium bromide) in PBS was added per well and cells were further incubated for 2 h at 37 °C. The medium was then discarded and cells were washed with 100 µl PBS before adding 100 µl 2-propanol/10 N HCl (250:1) in order to dissolve formazan granules. The absorbance at 570 nm was measured using a microplate reader (Tecan Infinite M200Pro), and cell viability was expressed as percentage relative to the respective methanol control. $IC_{50}$ values were determined by sigmoidal curve fitting.

## recA Reporter strain Bioluminescence assays

*DNA-damage M. smegmatis* mc²155 reporter strains (harboring P*recA*-LUX or P*radA*-LUX plasmids) were kindly supplied by Prof. Digby F Warner from the University of Cape Town, Institute of Infectious Disease & Molecular Medicine (IDM). Briefly, the reporter strains were grown to mid-log phase (optical density at 600 nm [$OD_{600}$] of 0.4 to 0.8) and diluted to a starting $OD_{600}$ of ~0.05 before inoculation into 96-well microtiter plates. Twofold serial dilutions of each test compound were prepared in white 96-well plates (Greiner) containing the reporter strain in a final volume of 150 µl per well, and then the plates were incubated in a 5% $CO_2$ humidified incubator at 37 °C for 1 d. Luminescence (recorded as relative luminescence units [RLU]) was measured every 3 h using a microplate reader (Tecan Infinite M200Pro) with a 1.0-s measurement interval time after a positioning delay of 0.2 s, and plates were shaken for 5 s at 120 rpm before reading. Ciprofloxacin served as the maximum luminescence control. All growth media contained kanamycin (20 µg/ml) to maintain the reporter constructs.

## Gene cloning, protein expression, and purification with general condition

The genes *aaproC, mypE, mypF, aaleuB, mypK* were amplified by PCR using the genomic DNA of *Actinoplanes awajinensis* ATCC 33919 as the template. The obtained fragments were subcloned into the pHISTEV or pET28b vector via Gibson assembly (NEB, E2611S) or conventional T4 ligation (ThermoFisher Scientific, EL0014), resulting in the protein expression plasmids. The genes *aaleuC* and *mypL_N* were also amplified by PCR using the genomic DNA of *Actinoplanes awajinensis* ATCC 33919 as a template. The obtained fragments were subcloned into the pACYCDuet-1 vector via conventional ligation stepwise, resulting in the expression plasmid pACYCDuet-1-*AaleuCmypL_N*.

One liter of 2YT medium (16 g tryptone, 10 g yeast extract, and 5 g NaCl per liter) supplemented with kanamycin was inoculated with *E. coli* BL21 (DE3) strains that harbored expression plasmids accordingly. The culture was incubated with shaking at 37 °C, 200 rpm until $OD_{600}$ reached 0.6. The culture was then transferred into a 16 °C shaker (Infors) for 30 min. Once the culture was cooled to 16 °C, the expression was induced by the addition of 0.1 mM Isopropyl-β-D-thiogalactopyranosid (IPTG). After overnight shaking at 16 °C, cells were harvested by centrifugation and resuspended in lysis buffer [20 mM Tris-HCl (pH 8.0), 200 mM NaCl, 20 mM imidazole, 10% glycerol, and 1 mM DTT]. Lysis was achieved via passage through a cell disrupter (Constant Systems) at 30,000 psi, and cell debris was removed by centrifugation (19,500 × *g*, 4 °C, 60 min). The supernatant was collected and loaded onto a 5-mL Histrap HP column (GE healthcare, GE17-5248-01) pre-equilibrated with lysis buffer. The column was washed extensively with lysis buffer (20 column volumes) before the protein was eluted with elution buffer (lysis buffer with 250 mM imidazole). Fractions containing N-terminal His-tag protein were directly loaded onto HiPrep 26/10 desalting column (GE healthcare, GE17-5087-01) pre-equilibrated in desalting buffer [20 mM Tris-HCl (pH 8.0), 200 mM NaCl, 10% glycerol and 1 mM DTT]. The samples were collected, concentrated and flash frozen for storage at −80 °C.

## Cloning, expression and purification of MypL_C

The gene *mypL_C* was amplified by PCR using the genomic DNA of *Actinoplanes awajinensis* ATCC 33919 as a template. The obtained fragment was subcloned into the pHISTEV vector via conventional ligation, resulting in the expression plasmid pHISTEV-*mypL_C*.

1 L of 2YT medium supplemented with kanamycin and 1 µM pyridoxal 5′-phosphate (PLP) was inoculated with *E. coli* BL21 (DE3) harboring pHISTEV-m*ypL_C*. The culture was incubated at 37 °C with 200 rpm shaking until $OD_{600}$ reached 0.6. The culture was then transferred into a 16 °C shaker for 30 min. Once the culture was cool, expression of MypL_C was induced by the addition of 0.1 mM IPTG. After overnight shaking at 16 °C, cells were harvested by centrifugation and resuspended in lysis buffer [50 mM PBS (pH 8.0), 300 mM NaCl, 20 mM imidazole, 1 µM PLP, and 1 mM DTT]. Lysis was achieved via passage through a cell disrupter at 30,000 psi, and cell debris was removed by centrifugation (19,500 × *g*, 4 °C, 60 min). The supernatant was collected and directly loaded onto a 5 mL Histrap HP column pre-equilibrated with lysis buffer. The column was washed extensively with lysis buffer (20 column volumes) before the protein was eluted with elution buffer (lysis buffer with 250 mM imidazole). Fractions containing N-terminal His-tag MypL_C were directly loaded onto HiPrep 26/10 desalting column pre-equilibrated in desalting buffer (50 mM PBS (pH 8.0), 300 mM NaCl, 1 µM PLP, and 1 mM DTT). The sample was collected, concentrated and flash frozen for storage at −80 °C.

## In vitro biochemical analysis of MypK

The reaction mixture containing 0.5 mM 4-methyl-2-oxopentanoate (**1**) (or other α-keto acids including 2-oxohexanoate (**2**) and 3-methyl-2-oxopentanoate (**3**)), 0.5 mM acetyl-CoA, 50 mM NaCl, 10 mM $MgCl_2$, 0.1 mg/mL BSA, 0.5 mM DTT, 10 mM Tris buffer (pH 7.5) and 1 µM of MypK in a total volume of 50 µL was incubated at 30 °C for 1 hour before being quenched by addition of 100 µL methanol. Precipitated protein was removed by centrifugation, and the supernatant was directly applied to LC-MS analysis. For the analysis of the kinetic parameters of MypK reactions, 100 µL reaction mixture containing 2 mM acetyl-CoA, 10 mM $MgCl_2$, 50 mM potassium phosphate buffer (pH8.0), 3 µM of MypK, and 10–1000 µM α-keto acids were incubated at 30 °C. The initial velocity of the reaction was estimated by monitoring the increase in absorbance at 412 nm according to Ellman's assays[31].

## In vitro biochemical analysis of MypL_c

The reaction mixture containing 0.1 mM 4-methyl-2-oxo-hexanoic acid, 1 mM L-glutamate (or other amino acids), 50 mM NaCl, 10 mM $MgCl_2$, 0.1 mg/mL BSA, 0.5 mM DTT, 10 mM Tris buffer (pH 7.5), and 100 nm MypL_c in total volume of 50 µL was incubated at 30 °C for 1 h before being quenched by adding 100 µL methanol. Precipitated protein was removed by centrifugation, and the supernatant was directly applied to LC-MS analysis.

## Feeding experiment

2 mL of the seed culture in GYM medium (Glucose 4.0 g, yeast extract 4.0 g, malt extract 10.0 g, $CaCO_3$ 2.0 g, 1000 mL $H_2O$, pH 7.2) was transferred into 20 mL GYM medium in 100 mL flasks and incubated at 30 °C with shaking at 180 rpm. After 72 h, 96 h, and 120 h of incubation, 300 µL of L-Methionine-(*methyl*-¹³C) solution (Cambridge Isotopes, 10 mg/mL, dissolved in water, filtered, and stored at −20 °C) was added. In contrast, 300 µL sterilized water was supplemented in the negative control. After another incubation of 48 h, 1 mL XAD-16 was

added. The mixture was incubated overnight, and the cells and XAD-16 were harvested by centrifugation at 8157 × g and resuspended by 30 mL MeOH. After 2 h of incubation, the MeOH was evaporated, and the residues were redissolved in 1 mL MeOH for UHPLC-HRMS analysis.

## One pot in vitro reconstitution of homoleucine, homo-norleucine and homoisoleucine

The reaction mixture contained 1 mM **1, 2**, or **3**, 1 mM acetyl-CoA, 3 mM NAD[+], 10 mM MgCl$_2$, 0.5 mM MnCl$_2$, 0.2 mM DTT, 50 mM potassium phosphate buffer (pH 8) and 1 μM of MypK, 0.5 μM aaLeuC and MypL$_N$, and 2 μM aaLeuB in total volume of 50 μL. After overnight incubation at 30 °C, 10 μM PLP, 2 mM L-glutamate, 0.5 μM aaLeuC and MypL$_N$, 2 μM aaLeuB and 1 μM MypLc were added. The reaction was quenched by adding 100 μL methanol after 2 hours of incubation at 30°C. Precipitated protein was removed by centrifugation, and the supernatant was transferred to a new tube and then dried under nitrogen. The residual material was dissolved in 5 μL MilliQ water. 2 μL NaHCO$_3$, 8 μL acetone, and 2 μL of 1 % D-FDLA were added to the solution. The derivatization was carried out at 37 °C for 1 h with rotation. The reaction was stopped by adding 33 μL 1 N HCl. The mixture was further mixed with 50 μL acetonitrile and analyzed by LC-MS.

## In vitro analysis of MypE

The reaction mixture containing 0.5 mM substrate, 1 mM α-ketoglutarate, 20 μM FeSO$_4$, 0.2 mM ascorbate, 0.5 mM DTT, 5 mM potassium phosphate buffer (pH8.0) and 3 μM of MypE in a total volume of 50 μL was incubated at 30 °C for 3 h before being quenched by the addition of 50 μL of methanol. Precipitated protein was removed by centrifugation, and the supernatant was transferred to a new tube. The supernatant was removed under a stream of nitrogen and the residual material was dissolved in 5 μL MilliQ water. 2 μL NaHCO$_3$, 8 μL acetone and 2 μL of 1% D-FDLA were added into the solution[4]. The derivatization was carried out at 37 °C for 1 h with rotation. The reaction was stopped by adding 33 μL 1 N HCl. The mixture was further mixed with 50 μL acetonitrile and analyzed by LC-MS.

## Production of 4-hydroxymethylhomoisoleucine in *E. coli* BL21 (DE3) harboring pET28b-MypE

In total, 100 mL of 2YT medium containing kanamycin were inoculated with *E. coli* BL21 (DE3) harboring pET28-*mypE* and incubated at 37°C until OD$_{600}$ reached 0.6. 0.1 mM IPTG was added to induce the expression of MypE. The bacteria were further incubated with agitation (200 rpm) at 16 °C overnight. The cells were harvested by centrifugation and resuspended in 100 mL M9 minimal medium containing 2% glucose and 1 mM homoisoleucine. The reaction was carried out for 2 days and cells were removed by centrifugation. The supernatant was transferred to a new Falcon tube. Water was removed by lyophilization. After the resuspension in methanol and centrifugation, the residual material was removed. 4-hydroxymethylhomoisoleucine was further purified using semi-preparative UHPLC equipped with a Luna 5 μm C18 (250 × 10 mm, 100 Å, Phenomenex) using liner water-acetonitrile (0.1% formic acid) gradient.

## In vitro analysis of MypF

The reaction mixture containing 0.5 mM 4-hydroxymethylhomoisoleucine, which was isolated from *E. coli* BL21 (DE3) overexpressing MypF, 1 mM ZnSO$_4$, 2 mM NAD[+], 0.5 mM DTT, 50 mM of glycine buffer (pH 10) and 5 μM of MypF in a total volume of 50 μL was incubated at 30 °C for 12 h before being quenched by adding 100 μL methanol. Precipitated protein was removed by centrifugation, and the supernatant was directly applied to LC-MS analysis. NADH formation in the reaction was monitored via the increase in absorbance at 340 nm.

## Coupled assays of 4-ethylproline formation

The reaction mixture containing 0.5 mM homoisoleucine, 1 mM α-ketoglutarate (Sigma-Aldrich, 75890), 20 μM FeSO$_4$, 0.2 mM ascorbate, 0.5 mM DTT, 5 mM potassium phosphate buffer (pH8.0) and 3 μM of MypE in a total volume of 100 μL was incubated at 30 °C for 1 h. The incubation proceeded after the solution was then supplemented with glycine buffer (pH 10), ZnCl$_2$, NAD[+], MypF and aaProC to a final concentration of 100 mM, 1 mM, 2 mM, 5 μM, and 5 μM respectively overnight at 30 °C. The reaction was quenched by adding 100 μL methanol. Precipitated protein was removed by centrifugation and the supernatant was transferred to a new tube and dried by evaporation under nitrogen. Residual material was dissolved in 5 μL MilliQ water. 2 μL NaHCO$_3$, 8 μL acetone and 2 μL of 1% D-FDLA were added into the solution. The derivatization was carried out at 37 °C for 1 hour with rotation. Reaction was stopped by adding 33 μL 1 N HCl. The mixture was further mixed with 50 μL acetonitrile and analyzed by LC-MS.

## Intact protein measurements

All ESI-UPLC-HRMS measurements were performed on a Dionex (Germering, Germany) Ultimate 3000 RSLC system coupled with a maXis4G Q-ToF mass spectrometer using an Aeris Widepore XB-C8 column (3.6 μm, 150 × 2.1 mm, Phenomenex) column. Separation of 1 μL sample was achieved by a multistep gradient from (A) H$_2$O + 0.1 % FA to (B) ACN + 0.1 % FA at a flow rate of 300 μL/min and 45 °C. Chromatographic conditions were as follows: 0–1 min, 2 % B; 1-11 min, 2–75 % B; 11–14 min, 75% B; 14-14.35 min, 75-2% B; 14.35-15.35 min, 2 % B. UV spectra were recorded by a DAD in the range from 200 to 600 nm. The LC flow was split to 75 μL/min before entering the maXis 4G hr-ToF mass spectrometer (Bruker Daltonik, Bremen, Germany) using the standard Bruker Apollo II ESI source. In the source region, the temperature was set to 200 °C, the capillary voltage was 4000 V, the dry-gas flow was 5.0 L/min and the nebulizer was set to 1.0 bar. Mass spectra were acquired in positive ionization mode ranging from 150 to 2500 *m/z* at 2 Hz scan rate. Protein masses were deconvoluted by using the Maximum Entropy algorithm in DataAnalysis 5.3 (Bruker Daltonik GmbH).

## MPs bind to DnaN in nanomolar affinity

GM and MPs A, B, D, and E were investigated regarding their binding affinity to *M. smegmatis* DnaN (msDnaN) using MST by evaluating the dissociation constant $K_D$. Experiments were performed according to the standard protocol by NanoTemper using a Monolith NT-115 Microscale Thermophoresis device. *M. smegmatis* DnaN was labeled using Monolith Series Protein Labeling Kit BLUE-NHS and diluted to 50 nM with MST buffer (20 mM Tris, 150 mM NaCl, 0.05% Tween-20) in the assay. All compounds were titrated in triplicates in an appropriate concentration range for proper dose-response curves with 16 serial dilution steps. Data evaluation was carried out by the software MO.Affinity Analysis v2.3 from NanoTemper and visualized by Prism.

## Production of ecDnaN protein

The gene encoding for ecDnaN was cloned into a modified pCOLA Duet-1 vector (Novagen) encoding for an N-terminal Strep-tag II and TEV-protease recognition site. The protein was expressed in *E. coli* BL21 (DE3) in ZYM-5052 auto-inducing medium[32] at 20 °C for 20–24 h. The cell pellet was resuspended in a buffer containing 20 mM HEPES/NaOH pH 7.5, 150 mM NaCl, one tablet of complete EDTA-free protease inhibitor cocktail (Roche) and lysed by sonication. The protein was isolated from the supernatant after centrifugation for 1 h at 100,000 × g using a self-packed 10 mL column with Strep-Tactin Superflow High Capacity resin (IBA) and eluted from the column with a single step of 5 mM D-desthiobiotin. The affinity tag was cleaved off with TEV protease (1:50 mg/mg) at 4 °C overnight. Gel filtration was carried out using a HiLoad 16/600 Superdex 200 pg column (GE Healthcare) in 20 mM HEPES/NaOH pH 7.5, 150 mM NaCl. The peak

fractions were concentrated and flash-frozen in liquid nitrogen for crystallization screening.

## Crystallization

Crystallization trials were set up at room temperature with a Crystal Gryphon crystallization robot (Art Robbins Instruments) in Intelli 96-3 plates (Art Robbins Instruments) with 200 nL protein solution at different concentrations and 200 nL reservoir solution. Crystals were obtained after a few days in co-crystallization setups (Conditions: Supplementary Table 9). Crystals were, after harvesting, cryo-protected by addition of 10% (v/v) (2 R,3 R)-2,3-butanediol.

## Data collection and processing

Data collection for ecDnaN with MP A was performed at beamline P11 at the Petra III storage ring (Deutsches Elektronen-Synchrotron, Hamburg, Germany)[33], for ecDnaN with GM at beamline BL 14.2 of the BESSY II (Helmholtz Zentrum Berlin, Germany)[34] and for ecDnaN with CGM at beamline X06DA (PXIII) of the Swiss Light Source (Paul Scherrer Institute, Villigen, Switzerland). All datasets were recorded at a temperature of 100 K. Data processing was carried out using the AutoPROC[35] toolbox (Global Phasing, v. 1.0.5) executing XDS (v. Jan 10, 2022)[36], Pointless (v. 1.12.14)[37], and Aimless (v. 0.7.9)[38].

## Structure determination, refinement, and model building

The structure of the ecDnaN ligand-complexes were determined by molecular replacement using the available structure from the PDB (4K3L[39]) as a search-model for Phaser (v. 2.8.3)[40] from the Phenix suite (v. 1.20.1-4487)[41]. The structural models were built using Coot (v. 0.9.6)[42] and crystallographic refinement was performed with Phenix.refine (v. 1.20.1-4487)[43] including the addition of hydrogens in riding positions and TLS-refinement. 5% of random reflections were flagged for the calculation of $R_{free}$. The model of ecDnaN in complex with MP A was at 2.1 Å resolution and refined to R/$R_{free}$ of 22/26% in space group C2. The structure of ecDnaN in complex with GM was at 1.7 Å resolution and refined to R/$R_{free}$ of 18/19% in space group C222. The structure of ecDnaN in complex with CGM was at 1.5 Å resolution and refined to R/$R_{free}$ of 17/18% in space group C222. Data collection and refinement statistics are summarized in Supplementary Table 8. Figures of crystal structures were prepared using the PyMOL Molecular Graphics System version 2.4.0 (Schrödinger, LLC).

## Reporting summary

Further information on research design is available in the Nature Portfolio Reporting Summary linked to this article.

## Data availability

The mycoplanecin biosynthetic gene cluster sequence generated in this study has been deposited in GenBank under accession code OR083095. The protein crystal structure data generated in this study have been deposited in the Protein Data Bank under PDB IDs 8CIX, 8CIY, and 8CIZ. All data that support the findings of this study are available in the main text and the supplementary information. Source data are provided with this paper.

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

## Acknowledgements

This work was supported in part by the Bill & Melinda Gates Foundation [INV-001913]. C.F. and R.M. thank the support from the Helmholtz International Lab (InterLabs-0007). Research in Rolf Müller's laboratory is funded by the Deutsche Forschungsgemeinschaft (DFG), the Bundesministerium für Bildung und Forschung (BMBF), and the Deutsches Zentrum für Infektionsforschung Standort Hannover-Braunschweig. Y.L. was partially supported by the "China and Germany Postdoctoral Exchange Program". The authors thank Stephanie Sans from Evotec for performing part of the bioactivity assays and Digby F Warner from the University of Cape Town for providing DNA-damage *M. smegmatis* mc²155 reporter strains. We thank DESY (Hamburg, Germany), the Helmholtz Center Berlin (HZB, Berlin, Germany) and the Paul Scherrer Institute (PSI, Villigen, Switzerland) for the provision of experimental facilities. Parts of this research were carried out at PETRA III (DESY) using beamline P11, at beamline BL 14.1 of the BESSYII electron storage ring (HZB) and at beamline X06DA at the SLS (PSI). The authors wish to thank the beamline staff for technical support during data collection. We thank Ute Widow, HZI, for excellent technical assistance in the wet lab.

## Author contributions
C.F. and Y.L. performed the gene cluster analysis and in vitro biochemical study. O.K. and C.F. performed and analyzed the SSN analysis. C.W. and C.B. performed compound isolation and structural elucidation experiments. P.L. performed the study of the co-crystal structure of DnaN and mycoplanecin A. C.W. performed the binding assay. S.R. performed the bioactivity measurements. M.N. contributed to fermentation for MP production. F.P.J.H. helped with results discussion. W.B. contributed to the protein crystallization data analysis. C.F. and R.M. wrote the paper with input from all authors. R.M. conceived the study. C.F., Y.L., C.W. and S. R. contributed equally to the manuscript. All authors read and approved the final manuscript.

## Funding

## Competing interests
The authors declare no competing interests.
