## [Peer Review File · Nature Communications]

Elucidation of unusual biosynthesis and DnaN-targeting mode of action of potent anti-tuberculosis antibiotics MycoplanecinREVIEWER COMMENTS

Reviewer #1 (Remarks to the Author):

The reviewer has been asked to comment on this submitted manuscript, mainly on the structural biology part. In this paper, the authors report the structural analysis of the complex of *E. coli* DnaN with the antibiotic mycoplasmanesin MP A, which is expected to have anti-tuberculosis activity, griselimycin GM and cyclohexyl GM CGM. Since it was not possible to obtain co-crystals of *Mycobacterium smegmatis* DnaN in complex with MP A, *E. coli* DnaN was used in this structural study. For comparison, the authors also determined the complex structure of GM or CGM with *E. coli* DnaN, which had previously been successfully analyzed by this group in complex structures with *Mycobacterium tuberculosis* and *Mycobacterium smegmatis* DnaNs (Science 2015).

The present crystal structures were determined with sufficient resolution to discuss the interactions in detail. Judging from Supplementary Table 7, the crystal structures were well refined and there would be no technical problems with this structural analysis. This structural analysis reveals the interaction of MPA with *E. coli* DnaN, but does not directly reveal the interaction of MPA with *Mycobacterium tuberculosis* DnaN. However, the information on the interaction of MPA is valuable.

The authors showed that mycoplanecins bind to DnaN with nanomolar affinity. Furthermore, the biosyntheses of the unusual components of MPs were reconstructed by eight enzymes *in vitro*. This work, including biosynthetic studies, bioactivity evaluation and validation of drug targets for mycoplanecin, would pave the way for further development against multidrug-resistant mycobacterial infections.

Several concerns are listed below.

The binding of these antibiotics to DnaN is discussed on the basis of the complex structures with *E. coli* DnaN. Therefore, the authors should show the comparison between *E. coli* DnaN and *Mycobacterium* DnaN so that we can see the conservation of amino acid residues in the binding pocket. In addition, the authors should show the structural comparison of the binding pockets of *E. coli* DnaN and *Mycobacterium* DnaN.

In *E. coli* DnaN, several studies have shown structural changes at M362 and S346 in the binding pocket upon partner binding. It would be better to describe whether there is a conformational change at M362 or S346 when the ligands bind to *E. coli* DnaN.

Regarding the superimposition of structures, the authors should describe in detail whether the superimpositions were performed on the whole structure or only on specific regions.

The labels A, B, etc. in some legends should probably be in lower case.

Other specific comments are listed below.

Line 172

Since the authors mention that they tested several Gram positive species, the authors should indicate the specific species being tested.

Line199-200

The resolution values for structural analysis should be the same as those in Supplementary Table 7.

Line 203

It should be mentioned that R365 of *E. coli* DnaN is conserved in *Mycobacterium tuberculosis* or *Mycobacterium smegmatis* DnaN.

Line 206

The authors described that the linear N-terminal parts of the ligands bind to subsite 2. It should first be mentioned that there are two interacting sites on the molecular surface of DnaN, called subsite 1

and subsite 2.

Line 207, 217-218

It would be easier to understand if the authors could indicate the direction in which the structures were tilted with arrows in the appropriate figures.

Line 210-222

The temperature factors indicate that the binding of residues 1-3, 4-5, and 10 of MPA to E. coli DnaN is strong and that the binding of residues 6-9 to E. coli DnaN is weak (Supplementary Figure 8). If the difference in ligand conformation is due to crystal packing, it would be difficult to determine the binding strength from the crystal structure. If the conformations of the ligands were changed by the crystal packing, the temperature factors should be smaller due to the packing interaction by the symmetric molecules. The reviewer thinks that the description here needs to be reconsidered.

Figure 3

It would be helpful to list the position numbers in the chemical formula.

Figure 4

180°C should read 180° in the legend.

It would be helpful if the authors could indicate where subsite 2 is located on the surface of the molecule. It would also be helpful if the authors could indicate Pro8.

Supplementary Figure 5

The electron density of MPA is too broad for a sigma level of 1.0 at 2.1 Å resolution. Any comments on this would be appreciated. If possible, the omit map could be shown to emphasize the validity of the structure.

The structures of the three DnaNs appear to be identical. Make sure that the authors have shown the DnaN corresponding to each ligand.

Supplemental Figure 6

These are very important figures showing the schematic diagrams of the interactions of the three ligands with the E. coli DnaN, but they should be enlarged because the chemical structures and letters are too small to see.

Supplementary Figure 7

180°C should read 180° in the legend.

The legend should clearly state whether the gray molecular surface is E. coli DnaN or Mycobacterium DnaN.

The position of Pro8 where differences in structure were observed should be shown. Position 6 should also be shown.

Supplementary Figure 8

The structures of the three DnaNs appear to be identical. Make sure that the authors have shown the DnaN corresponding to each ligand.

Supplementary Table 7

The lattice constant angle 90.00 must be 90 because it is strictly defined as 90.

Supplementary Table 9

The temperature of 273 K is not a common condition for protein crystallization. The authors mentioned that DnaN was crystallized at room temperature in the line 779. Check the crystallization conditions.

Reviewer #2 (Remarks to the Author):

This manuscript describes the characterization of the previously unknown biosynthetic gene cluster (BGC) of mycoplanecins (MP), which were discovered in the 1980s and had potent anti-mycobacterial activity, by genome mining using bait genes from the 4-methylproline pathway and a new derivative, Mycoplanecin E, which was identified by 1D-, 2D- nuclear magnetic resonance (NMR) experiments and Marfey's analysis in addition to known compounds Mycoplanecin A, B, D. It is worthy to point out that Mycoplanecin E had exhibited significantly improved pharmaceutical properties compared to GM. In addition, co-crystal structure of mycoplanecin A-bound DnaN was also solved for target validation. At last, their biosynthetic pathway was investigated and proposed, especially the unique structural features: the L-homoleucine, L-homonorleucine and (2S,4R)-4-ethylproline building blocks.

The issues about the manuscript are listed to be concerned as following :

1. It is very interesting that radical SAM enzyme MypH may be involved in 4-EtPro and 4-PrPro biosynthesis. Have you tried to test the enzymatic activity of MypH?
2. Fig 3a, the chemical structures have already displayed in Fig 2.
3. MPs and GM are very similar in their structures, while they have different affinity to msDnaN. It will be interested to explain more about the structure activity relation of these compounds based on the complex crystal structure.
4. In Fig. 6f, L-leucine and L-allo-isoleucine were observed to be completely converted into related products catalyzed by MypE which were detected just by LC-MS, have you tried to isolate the related products and analyze by NMR?
5. In Fig. 6e (II), what does the peak at RT ca.19.5 min stand for?
6. Suggest to add the detail structure elucidation of the new compound MP E.
7. The authors had detected the formation of 4-EtPro by using 12 with related enzymes in Fig.6e, have you test the formation of 4-PrPro?
8. In TEXT, there are two parts of references in P19 and P29, it is necessary to integrate into one with sole number.
9. In SI, Figure S6 is a little obscure, high-resolution figure is needed.

Reviewer #3 (Remarks to the Author):

The manuscript by Müller and coworkers describes the targeted discovery of the mycoplanecin (MP) biosynthetic pathway, the isolation, structural and biological evaluation of three known and one new MP analog, crystal structure analysis of MP A, the structurally related griselimycin (GM) and a GM analog in complex with their target DnaN, and the in vitro reconstitution of all biosynthetic enzymes involved in the construction of the non-proteinogenic amino acids found in the MP structure. The biosynthetic pathway was identified by genome mining using the genes encoding enzymes for the prototypical 4-methylproline residue in GM biosynthesis. Given GM's potent anti-tuberculosis antibiotic effects, this aimed to discover novel TB inhibitors. While the compound class of the MPs had been known since the 1980s, this work now identified the respective biosynthetic gene cluster for the first time, attesting to the feasibility of this targeted genome mining approach to find antibiotics active against TB. MPs show even enhanced activity compared to GMs, with the novel analog being the most potent congener. This effect probably results from the increased lipophilicity of MPs, particularly the novel analog. Müller et al. characterized the inhibitory effect of all compounds and nicely illustrated their binding mode to DnaN. In addition, the extensive study of the amino acid precursor biosynthetic routes by in vitro analysis of all respective recombinant enzymes is of high interest. Particularly, the elucidation of the 4-alkylproline pathway is of high interest, as it sets the basis for the development of 4-alkylproline biosynthetic cassettes that might, in the future, be used to incorporate this residue into other peptide frameworks to design novel anti-TB antibiotics. Overall, this paper contains a large set of

thoroughly conducted and important biosynthetic investigations and provides new insights into anti-TB antibiotic discovery and modes of action, along with an in-depth analysis of crucial biosynthetic steps. Thus, I am convinced this work will fascinate the natural product community and beyond. The manuscript is also very well written and illustrated and, therefore, a pleasure to read. I therefore recommend its acceptance virtually as is, with just a few very minor suggestions, mainly concerning the manuscript text:

- The introduction mentions that the Pro8 residue is metabolically unstable and prone to oxidation. For clarity, this position may be indicated in the GM structure in Figure 1C.
- Figure 1a shows the biosynthesis of 4-methylprolin but is hardly mentioned in the manuscript text. The authors should add a sentence on putative enzyme functions, which will also be investigated later in the manuscript.
- Similar for Figure 1b: please add 2-3 sentences to the depicted analysis.

Supporting Information:

- For some of the ¹³C NMR spectra, phasing should be improved (particularly Fig. S17)
- Please include the position numbering from the NMR data tables in the respective molecular structures of MPs
- The ¹³C NMR spectrum of MP E seems to be missing (the data is present in the table, though) -> please include
- The ¹³C NMR spectrum of MP D is borderline; however, as this is a known compound, this is still acceptable.

REVIEWER COMMENTS

Reviewer #1 (Remarks to the Author):

The reviewer has been asked to comment on this submitted manuscript, mainly on the structural biology part. In this paper, the authors report the structural analysis of the complex of *E. coli* DnaN with the antibiotic mycoplasmanesin MP A, which is expected to have anti-tuberculosis activity, griselimycin GM and cyclohexyl GM CGM. Since it was not possible to obtain co-crystals of *Mycobacterium smegmatis* DnaN in complex with MP A, *E. coli* DnaN was used in this structural study. For comparison, the authors also determined the complex structure of GM or CGM with *E. coli* DnaN, which had previously been successfully analyzed by this group in complex structures with *Mycobacterium tuberculosis* and *Mycobacterium smegmatis* DnaNs (Science 2015).

The present crystal structures were determined with sufficient resolution to discuss the interactions in detail. Judging from Supplementary Table 7, the crystal structures were well refined and there would be no technical problems with this structural analysis. This structural analysis reveals the interaction of MPA with *E. coli* DnaN, but does not directly reveal the interaction of MPA with *Mycobacterium tuberculosis* DnaN. However, the information on the interaction of MPA is valuable. The authors showed that mycoplanecins bind to DnaN with nanomolar affinity. Furthermore, the biosyntheses of the unusual components of MPs were reconstructed by eight enzymes *in vitro*. This work, including biosynthetic studies, bioactivity evaluation and validation of drug targets for mycoplanecin, would pave the way for further development against multidrug-resistant mycobacterial infections.

Several concerns are listed below.

The binding of these antibiotics to DnaN is discussed on the basis of the complex structures with *E. coli* DnaN. Therefore, the authors should show the comparison between *E. coli* DnaN and *Mycobacterium* DnaN so that we can see the conservation of amino acid residues in the binding pocket. In addition, the authors should show the structural comparison of the binding pockets of *E. coli* DnaN and *Mycobacterium* DnaN.

Answer: An additional figure containing a sequence alignment of E. coli, M. tuberculosis and M. smegmatis DnaN and a binding site comparison (E. coli, M. tuberculosis) regarding the amino acid conservation has been added as the new Supplementary Figure 5.

In *E. coli* DnaN, several studies have shown structural changes at M362 and S346 in the binding pocket upon partner binding. It would be better to describe whether there is a conformational change at M362 or S346 when the ligands bind to *E. coli* DnaN.

Answer: We have added the sentence “Compared to available structures of uncomplexed DnaN, only a few residues undergo significant conformational changes upon binding of the compounds - most notably Met362, which has been described to function as a gate between the subsites.” to the structure description. A detailed analysis of conformational changes in the binding pocket have been added as a new Supplementary Figure.

Regarding the superimposition of structures, the authors should describe in detail whether the

superimpositions were performed on the whole structure or only on specific regions.

Answer: In principle, all superimpositions were done on the whole structure level. In the cases where the asymmetric unit contained more than a single chain of DnaN (e.g. a ring), the superimposition was done on a single-chain level (monomeric DnaN), due to PyMol's algorithm having problems with superimposing multimers correctly.

The description of the superimposition has been added to the corresponding figure legends, including Figure 4 and Supplementary Figure 8.

The labels A, B, etc. in some legends should probably be in lower case.

Answer: Upper case labels have been corrected.

Other specific comments are listed below.

Line 172

Since the authors mention that they tested several Gram positive species, the authors should indicate the specific species being tested.

Answer: Thanks for this helpful comment. We have indicated the names of bacterial species we have tested against mycoplanecins but showed no activity. We modified the corresponding text to "MPs were additionally tested against a panel of Gram-positive (*S. aureus* Newman and *B. subtilis* DSM 10) and Gram-negative (*E. coli* BW25113, *C. freundii* DSM 30039 and *A. baumannii* DSM 30008), but exhibited no activity up to 64 ug/mL. The mycobacteria-specific inhibitory bioactivity of MPs highlighted the high selectivity of this compound class, also concerning members of the human microbiota that are likely spared in the potential use of MP in TB therapy."

Line199-200

The resolution values for structural analysis should be the same as those in Supplementary Table 7.

Answer: The three-digit resolutions in the table are the result of the resolution shell determination by the processing software (resolution shells are calculated based on the numbers of included reflections). The refinement software rounded those values to two-digit already. We had then rounded these values to in our eyes more appropriate single-digit numbers. We'll now use the two-digit values as used in the refinement.

Line 203

It should be mentioned that R365 of *E. coli* DnaN is conserved in *Mycobacterium tuberculosis* or *Mycobacterium smegmatis* DnaN.

Answer: We appreciate this suggestion. This has been added accordingly. The corresponding text has been modified to "...but the additional oxygen in the N-terminal α -keto butyrate residue of MP A provides a second hydrogen bond acceptor for Arg365 (Supplementary Fig. 6), which is also conserved in mycobacterial DnaN."

Line 206

The authors described that the linear N-terminal parts of the ligands bind to subsite 2. It should first be mentioned that there are two interacting sites on the molecular surface of DnaN, called subsite 1

and subsite 2.

Answer: We appreciate this suggestion. An explanation has been added as "The binding pocket can be divided into a larger (subsite 1) and a smaller cavity (subsite 2) and as has been shown previously for GM and mycobacterial DnaN, all compounds occupy both subsites of the binding pocket."

Line 207, 217-218

It would be easier to understand if the authors could indicate the direction in which the structures were tilted with arrows in the appropriate figures.

Answer: The tilt indicator has been added to the new Supplementary Figure 8 (previous Supplementary Figure 7).

Line 210-222

The temperature factors indicate that the binding of residues 1-3, 4-5, and 10 of MPA to E. coli DnaN is strong and that the binding of residues 6-9 to E. coli DnaN is weak (Supplementary Figure 8). If the difference in ligand conformation is due to crystal packing, it would be difficult to determine the binding strength from the crystal structure. If the conformations of the ligands were changed by the crystal packing, the temperature factors should be smaller due to the packing interaction by the symmetric molecules. The reviewer thinks that the description here needs to be reconsidered.

Answer: We did not intend to use the B-factors to directly compare the binding-strength between the ligands and have rephrased the corresponding passage to "Although differences in crystal packing make a direct comparison of the binding strength difficult, the linear part of the ligands (1-3) and the residues adjacent to the bifurcation (4-5, 10) seem to be bound tighter/show less flexibility than the opposing residues (6-9) in all structures of DnaN/GM complexes..."

The less tight bound Pro8-region is indeed involved in a crystal contact in the ecDnaN structures, resulting in lower relative B-factors in this area as compared to the mycobacterial co-structures. However, the general pattern of highest B-factors at Pro8 and lowest around the ligand's bifurcation site is not altered. Also, the ligands in the ecDnaN co-structures are rather tilted out of the binding site (as compared to mycobacterial DnaNs); one would expect the opposite from a crystal contact in this region. We do thus not expect the crystal contacts to have a major influence on the ligands' conformations.

Figure 3

It would be helpful to list the position numbers in the chemical formula.

Answer: We appreciate this helpful comment. We have indicated all residues of MPs with position numbering in Figure 3a.

Figure 4

180°C should read 180° in the legend.

It would be helpful if the authors could indicate where subsite 2 is located on the surface of the molecule. It would also be helpful if the authors could indicate Pro8.

Answer: The legend of Figure 4 has been corrected. Subsites and indicators for residues 6 and 8 have been added to the figure.

Supplementary Figure 5

The electron density of MPA is too broad for a sigma level of 1.0 at 2.1 Å resolution. Any comments on this would be appreciated. If possible, the omit map could be shown to emphasize the validity of the structure.

Answer: A new Supplementary Figure 6 (previous Supplementary Figure 5) including omit maps (from the Phenix composite omit map tool) has been added. Unfortunately, the electron density for MPA in the ecDnaN co-structure does not show more details. The B-factors of this structure are generally relatively high resulting in less detailed electron density maps than usually to be expected for this resolution. This is an intrinsic property of this very sample, but this is the best co-crystal of MPA and ecDnaN we were able to generate.

The structures of the three DnaNs appear to be identical. Make sure that the authors have shown the DnaN corresponding to each ligand.

Answer: The cartoon representations of the underlying DnaNs have been corrected in the new Supplementary Figure 6 (previous Supplementary Figure 5).

Supplemental Figure 6

These are very important figures showing the schematic diagrams of the interactions of the three ligands with the E. coli DnaN, but they should be enlarged because the chemical structures and letters are too small to see.

Answer: A new high-resolution figure (current Supplementary Figure 7) has been prepared and replaced the old one (previous Supplementary Figure 6).

Supplementary Figure 7

180°C should read 180° in the legend.

The legend should clearly state whether the gray molecular surface is E. coli DnaN or Mycobacterium DnaN.

Answer: The current Supplementary Figure 8 (previous Supplementary Figure 7) has been updated, and the legend has been corrected.

The position of Pro8 where differences in structure were observed should be shown. Position 6 should also be shown.

Answer: Tilt indicator has been added in the current Supplementary Figure 8 (previous Supplementary Figure 7).

Supplementary Figure 8

The structures of the three DnaNs appear to be identical. Make sure that the authors have shown the DnaN corresponding to each ligand.

Answer: The current Supplementary Figure 9 (previously Supplementary Figure 8) has been updated accordingly.

Supplementary Table 7

The lattice constant angle 90.00 must be 90 because it is strictly defined as 90.

Answer: This has been corrected in Supplementary Table 7.

Supplementary Table 9

The temperature of 273 K is not a common condition for protein crystallization. The authors mentioned that DnaN was crystallized at room temperature in the line 779. Check the crystallization conditions.

Answer: This was a typo and has been corrected to 293 K (20°C). Also, we swap Supplementary Table 8 and Supplementary Table 9 to make the two tables regarding protein crystallization stay together.

Reviewer #2 (Remarks to the Author):

This manuscript describes the characterization of the previously unknown biosynthetic gene cluster (BGC) of mycoplanecins (MP), which were discovered in the 1980s and had potent anti-mycobacterial activity, by genome mining using bait genes from the 4-methylproline pathway and a new derivative, Mycoplanecin E, which was identified by 1D-, 2D- nuclear magnetic resonance (NMR) experiments and Marfey's analysis in addition to known compounds Mycoplanecin A, B, D. It is worthy to point out that Mycoplanecin E had exhibited significantly improved pharmaceutical properties compared to GM. In addition, co-crystal structure of mycoplanecin A-bound DnaN was also solved for target validation. At last, their biosynthetic pathway was investigated and proposed, especially the unique structural features: the L-homoleucine, L-homonorleucine and (2S,4R)-4-ethylproline building blocks.

The issues about the manuscript are listed to be concerned as following :

It is very interesting that radical SAM enzyme MypH may be involved in 4-EtPro and 4-PrPro biosynthesis. Have you tried to test the enzymatic activity of MypH?

Answer: We appreciate this comment from reviewer 2. The radical SAM enzyme MypH is potentially involved in the biosynthesis of mycoplanecins. To figure out the function of MypH in mycoplanecin biosynthesis, we indeed tried to purify MypH to prove our assumption. However, after multiple attempts, MypH still remained untamed due to the notorious difficulty of expressing functional radical SAM protein. Hence, we switched to an alternative approach using stable isotope-labeled substrate in the feeding study. As depicted in Figure 6a, the 1 Da increase in 4-EtPro and 2 Da increase in 4-PrPro due to the supplementation of L-Methionine-(methyl-¹³C) suggested the important role of MypH in 4-EtPro and 4-PrPro biosynthesis. Our future work will focus on MypH, but we must first reconstitute a functional MypH.

Fig 3a, the chemical structures have already displayed in Fig 2.

Answer: We appreciate this comment from reviewer 2. We presented the chemical structures of MPs in both Fig 2 and Fig 3 to make the results more straightforward and easier to understand for the readers so they do not need to go back and forth between figures and pages. We intend to keep both also because they play different functions in different figures for different purposes. In Fig 2, the MP structures are mainly presented to help understand how the NRPS biosynthetic assembly line produces the peptides and what are the final products of the NRPS pathway. In Fig 3, the MP

structures are used to help readers compare and analyze the different bioactivities and binding affinities of MPs with different structures. To differentiate the MP structures in Fig 2 and Fig 3a, we have added indications for residues and position numbering in the MP structures in Fig 3a.

MPs and GM are very similar in their structures, while they have different affinity to msDnaN. It will be interested to explain more about the structure activity relation of these compounds based on the complex crystal structure.

Answer: A possible explanation has been added to the discussion: In comparison to GM and CGM, the higher potency of MP A might be attributed to the increased hydrophobicity of the substance and the larger van der Waals interface with DnaN. The 5-methyl-L-norleucine at position 6 might also play a role by pushing the generally less tight bound region around Pro8 out of the binding pocket but stabilizing the tighter interaction around the ligand's bifurcation.

In Fig. 6f, L-leucine and L-allo-isoleucine were observed to be completely converted into related products catalyzed by MypE which were detected just by LC-MS, have you tried to isolate the related products and analyze by NMR?

Answer: We appreciate this comment from reviewer 2. Several shreds of evidence presented in this paper, including the successful *in vitro* reconstitution of 4-EtPro using homoisoleucine, hydroxylation of homoisoleucine, L-leucine or L-allo-isoleucine by MypE, and the observation of MypE failed to hydroxylate L-valine, all suggested MypE is a specific 5-methyl hydroxylase. Due to all these observations, we did not isolate the product from MypE incubated either with L-leucine or L-allo-isoleucine.

In Fig. 6e (II), what does the peak at RT ca.19.5 min stand for?

Answer: The peak at RT ca.19.5 min corresponds to Marfey's derivatization product of N-methyl-D-leucine residue from the hydrolyzed MP D, which owns the same mass as L-homoleucine. However, the two derivatized molecules have different retention times. We now indicate this peak in the legend: The peak at ~19.5 min (retention time) in II corresponds to Marfey's derivatization product of N-methyl-D-leucine.

Suggest to add the detail structure elucidation of the new compound MP E.

Answer: We appreciate this helpful suggestion from reviewer 2. We now add a comprehensive description regarding the structure elucidation of MP E in the paragraph "Isolation and structure elucidation of MPs with unusual building blocks".

The authors had detected the formation of 4-EtPro by using 12 with related enzymes in Fig.6e, have you test the formation of 4-PrPro?

Answer: We appreciate this comment from reviewer 2. As we explained in answer to the first question from reviewer 2, the expression of radical SAM enzyme MypH in *E. coli* is still problematic due to the notorious difficulty of expressing functional radical SAM protein. After multiple attempts, MypH still remained untamed. Therefore, we switched to an alternative approach using stable isotope-labeled substrate in the feeding study. The 1 Da increase in 4-EtPro and 2 Da

increase in 4-PrPro due to the supplementation of L-Methionine-(*methyl*-¹³C) suggested the important role of MypH in 4-EtPro and 4-PrPro biosynthesis. Our future work will focus on MypH, but we must first reconstitute a functional MypH. The 4-PrPro reconstitution will have high priority in our following work program. We also apologize that several typos were found in Fig. 6e and have now been corrected in the new Fig. 6.

In TEXT, there are two parts of references in P19 and P29, it is necessary to integrate into one with sole number.

Answer: We appreciate this helpful comment. We now merged the previous two reference lists and placed the new one at the end of the manuscript.

In SI, Figure S6 is a little obscure, high-resolution figure is needed.

Answer: We appreciate this helpful suggestion. A new high-resolution figure has been prepared. We now increase the resolution of current Supplementary Figure 7 (previously Supplementary Figure 6) and believe the new figure can provide enough clarity.

Reviewer #3 (Remarks to the Author):

The manuscript by Müller and coworkers describes the targeted discovery of the mycoplanecin (MP) biosynthetic pathway, the isolation, structural and biological evaluation of three known and one new MP analog, crystal structure analysis of MP A, the structurally related griselimycin (GM) and a GM analog in complex with their target DnaN, and the in vitro reconstitution of all biosynthetic enzymes involved in the construction of the non-proteinogenic amino acids found in the MP structure. The biosynthetic pathway was identified by genome mining using the genes encoding enzymes for the prototypical 4-methylproline residue in GM biosynthesis. Given GM's potent anti-tuberculosis antibiotic effects, this aimed to discover novel TB inhibitors. While the compound class of the MPs had been known since the 1980s, this work now identified the respective biosynthetic gene cluster for the first time, attesting to the feasibility of this targeted genome mining approach to find antibiotics active against TB. MPs show even enhanced activity compared to GMs, with the novel analog being the most potent congener. This effect probably results from the increased lipophilicity of MPs, particularly the novel analog. Müller et al. characterized the inhibitory effect of all compounds and nicely illustrated their binding mode to DnaN. In addition, the extensive study of the amino acid precursor biosynthetic routes by in vitro analysis of all respective recombinant enzymes is of high interest. Particularly, the elucidation of the 4-alkylproline pathway is of high interest, as it sets the basis for the development of 4-alkylproline biosynthetic cassettes that might, in the future, be used to incorporate this residue into other peptide frameworks to design novel anti-TB antibiotics. Overall, this paper contains a large set of thoroughly conducted and important biosynthetic investigations and provides new insights into anti-TB antibiotic discovery and modes of action, along with an in-depth analysis of crucial biosynthetic steps. Thus, I am convinced this

work will fascinate the natural product community and beyond. The manuscript is also very well written and illustrated and, therefore, a pleasure to read. I therefore recommend its acceptance virtually as is, with just a few very minor suggestions, mainly concerning the manuscript text:

The introduction mentions that the Pro8 residue is metabolically unstable and prone to oxidation. For clarity, this position may be indicated in the GM structure in Figure 1C.

Answer: We appreciate this helpful suggestion from reviewer 3. We have marked and highlighted the Pro8 residue in the GM structure in Figure 1c.

Figure 1a shows the biosynthesis of 4-methylprolin but is hardly mentioned in the manuscript text. The authors should add a sentence on putative enzyme functions, which will also be investigated later in the manuscript.

Answer: We appreciate this helpful comment from reviewer 3. We now replaced the previous sentence "The dioxygenase GriE and the dehydrogenase GriF are essential for (2*S*,4*R*)-4-MePro biosynthesis in the GM pathway (Fig. 1a)." with a detailed explanation on the 4-MePro pathway as "In our previous study, the Fe(II)/ α -ketoglutarate (α -KG)-dependent dioxygenase GriE was proven to initiate the biosynthesis of (2*S*,4*R*)-4-MePro by hydroxylating L-leucine to form (2*S*,4*R*)-5-hydroxyleucine. The zinc-dependent dehydrogenase GriF then converts (2*S*,4*R*)-5-hydroxyleucine to (2*S*,4*R*)-4-methylglutamate-5-semialdehyde. The resulting semialdehyde will be turned to (2*S*,4*R*)-4-MePro by a spontaneous cyclization and a final reduction by the pyrroline-5-carboxylate reductase ProC or GriH (Fig. 1a).".

Similar for Figure 1b: please add 2-3 sentences to the depicted analysis.

Answer: We appreciate this suggestion from reviewer 3. The analysis of Fig 1b was actually blended across the whole first paragraph of the results section. To increase the clarity, we refer to Figure 1b in several new places in the text of the first paragraph in the results. We also modify the sentence "One hit similar to the GM pathway drew our attention because it exhibited significant differences." to "Among the unknown hits, which are mainly nonribosomal peptide synthetase (NRPS) or NRPS hybrid BGCs, one hit similar to the GM pathway drew our attention because it exhibited significant differences."

Supporting Information:

- For some of the ¹³C NMR spectra, phasing should be improved (particularly Fig. S17)

Answer: Thanks for this helpful comment. We now improved the phasing for the ¹³C NMR spectra.

- Please include the position numbering from the NMR data tables in the respective molecular structures of MPs

Answer: As suggested, we replaced the chemical structures of MPs with position numbering related to the NMR data tables.

- The ¹³C NMR spectrum of MP E seems to be missing (the data is present in the table, though) -> please include

Answer: Due to the amount issue and insensitivity of ¹³C NMR measurement, we were unable to measure the ¹³C NMR spectra of MP E with desirable quality. The ¹³C NMR data presented in

Supplementary Table 13 was actually summarized from the 2D NMR spectra of MP E.

- The ^{13}C NMR spectrum of MP D is borderline; however, as this is a known compound, this is still acceptable.

Answer: We appreciate this comment from reviewer 3. As we explained in the answer to the previous question about ^{13}C NMR spectrum of MP E, we also experienced the compound amount issue for MP D. We still managed to record the ^{13}C NMR spectrum of MP D with acceptable quality. However, the quality of this spectrum is certainly not superior as reviewer 3 pointed out.

REVIEWERS' COMMENTS

Reviewer #1 (Remarks to the Author):

The authors have considered all comments and concerns from the reviewer #1 and have carefully revised their original manuscript. They have also provided sufficient explanation and, therefore, there are no crucial comments or concerns from the reviewer #1.

The reviewer #1 found one typo on line 241.

Incorrect: difficultthe

Correct: difficult, the

Reviewer #2 (Remarks to the Author):

I think the authors have responded well to the reviewers comments. I favor publication in Nature Communications.

Reviewer #3 (Remarks to the Author):

Müller and coworkers submitted the revised version of their manuscript 'Mycoplanecins, potent anti-tuberculosis antibiotics: Elucidation of unusual biosynthesis and DnaN-targeting mode of action'. The manuscript was already of high quality in the initial submission. The authors have now addressed all comments by the reviewers in a sufficient way. I therefore suggest acceptance of this nice work.